# Private and Stable Test-Time Adaptation with Differential Privacy

**Zefeng Li** [* 1 2]   **Qiaoyue Tang** [* 1]   **Mathias Lécuyer** [† 1]   **Evan Shelhamer** [† 1 2]

## Abstract

Test-time adaptation (TTA) can reduce error on new and different data by updating the model on these inputs during inference. However, these updates raise the issue of privacy w.r.t. the testing data, because the model parameters now depend on all past inputs. To control this privacy risk, we cast multiple popular TTA methods (Tent, EATA, SAR, DeYO, and COME) into differential privacy (DP) forms that apply per-sample gradient clipping and Gaussian noise for all updates. On ImageNet-C, our DP-TTA methods provide adequate privacy at small cost to accuracy, and in the low-privacy regime the clipping mechanism of DP can even improve the accuracy and stability of adaptation in the continual setting. These improvements to privacy and accuracy come at only modest computational overhead. These first results on private TTA raise awareness of the issue, inform the development of more private test-time updates, and identify per-sample clipping as an effective technique for improving the accuracy and stability of adaptation.

## 1. Introduction: Shifts, Updates, and Leaks

Distribution shifts that change the data during the deployment of a deep network can severely reduce the performance of predictions (Hendrycks & Dietterich, 2019; Geirhos et al., 2018; Yao et al., 2022). Such changes are common and include sensor noise due to extended use, unexpected weather and illumination changes, geographic or demographic shifts, and evolving data collection pipelines. *Test-time adaptation* (TTA) addresses these challenges by updating the model *during inference* using only unlabeled test data, without access to the original training set. A representative method, *Tent*, adapts by minimizing prediction entropy through updating only a sparse set of parameters (the affine parameters

[1]University of British Columbia [2]Vector Institute. Correspondence to: Zefeng Li <zefengli@cs.ubc.ca>, Qiaoyue Tang <qiaoyuet@cs.ubc.ca>.

*Proceedings of the 43rd International Conference on Machine Learning*, Seoul, South Korea. PMLR 306, 2026. Copyright 2026 by the author(s).

of normalization layers) to reduce error and preserve efficiency (Wang et al., 2021). Building on this, later methods introduce additional mechanisms to improve robustness and the stability of adaptation in realistic streaming conditions. For example, *EATA* reduces catastrophic forgetting and improves sample efficiency through selective updates and regularization (Niu et al., 2022). *SAR* improves robustness for small batch sizes and on mixed shifts by filtering unreliable updates and favoring stable optimization trajectories (Niu et al., 2023). *DeYO* argues that entropy can be an unreliable measure of uncertainty and proposes filtering and reweighting to mitigate the accumulation of noisy updates (Lee et al., 2024). *COME* further models uncertainty (via the Dirichlet distribution) to reduce overconfidence (Zhang et al., 2024).

Although TTA methods reduce error, they have overlooked potential side effects of their updates. We highlight for the first time the issue of *privacy*. Test data can contain sensitive information (e.g., medical images, faces, location data), and TTA incorporates this information in parameter updates. Once updated, the adapted model or its outputs may be inspected, queried, or shared, potentially leaking information about specific test data. Indeed, there is extensive evidence that learned models can leak information about data they were optimized on *during training*, enabling attacks such as membership inference (Shokri et al., 2017; Carlini et al., 2022) or reconstruction (Balle et al., 2022; Nasr et al., 2023; Carlini et al., 2021), including from model updates (Yin et al., 2021). In short, TTA raises the same privacy risks *during testing*, and so far these new risks have gone unnoticed by existing work on test-time updates.

*Differential Privacy* (DP) offers a principled way to bound leakage through models and their outputs, preventing membership and reconstruction attacks. The standard workhorse in DP deep learning is *DP-SGD*, which clips per-example gradients and injects calibrated Gaussian noise, enabling privacy accounting over the optimization procedure (Abadi et al., 2016). However, the direct application of DP-SGD to TTA is non-trivial. TTA is executed at small batch sizes (including a batch size of one), and many effective TTA methods rely on *data-dependent filtering* (e.g., entropy-/gradient-/consistency-based selection of updates) and *dynamic reweighting*, which change the effective updates across steps. Naïve implementations of these parts can impair stability and privacy.

We adapt the per-example gradient clipping and noise injection of DP-SGD to TTA methods to make their updates private—including **Tent**, **EATA**, **SAR**, **DeYO**, and **COME**—while preserving most of their specifics and accuracy-improving refinements. We introduce adjustments to privacy accounting and algorithms to make common TTA ingredients respect DP, including selective updates, streaming evaluation, and conservative uncertainty modeling. Our framework provides a shared lens for studying the *privacy/accuracy trade-off*: how much accuracy from adaptation can be retained under a given privacy budget, and which TTA design choices are more DP-friendly in practice. Surprisingly, we find that per-sample gradient clipping alone is an efficient mechanism to improve performance in TTA methods, leading to a $0.1\%$-*4.1% increase across multiple methods*, showing that clipping is useful in the no/low privacy regimes. Increasing DP noise then improves privacy guarantees, at the cost of lower accuracy. In summary, we make the following contributions:

- **DP-TTA:** We motivate, develop, and analyze DP versions of TTA methods to protect the privacy of test data. We build on DP-SGD, and adapt it to address practical issues when combining DP with specific TTA mechanisms.
- **Per-sample clipping for TTA:** We show that per-sample clipping is a general mechanism that improves performance across TTA methods. This per-sample clipping, crucial to privacy analysis, is of independent interest for TTA.
- **Empirical study of the privacy/accuracy trade-off:** We evaluate the impact of DP designs on the privacy/accuracy trade-off of TTA methods, and show that the cost of privacy is reasonable in the episodic and continual settings, especially due to improvements from per-sample clipping.

## 2. Preliminaries and Related Work

As the first to study private test-time adaptation, we review the test-time updates and differential privacy mechanisms necessary for our extensions and experiments.

### 2.1. Test-Time Adaptation

We consider a model $f_\theta : \mathcal{X} \to \mathcal{Y}$ pretrained on labeled source data $\mathcal{D}_{\text{tr}} = \{(x_i, y_i)\}$, resulting in parameters $\theta_0$. During deployment, inputs are drawn from a shifted test distribution and arrive as continual stream $\{x_i\}$ that are batched in a sequence $\{B_t\}_{t=1}^T$. Distribution shifts either happen during known episodes, or at unknown points of the data stream. Test-time adaptation (TTA) updates model parameters using test inputs (without labels) only, aiming to improve performance under distribution shift.

Most TTA methods rely on gradient optimization to minimize a test-time loss over batches and adapt the model parameters (in whole or in part) as $\theta_{t+1} = \theta_t - \eta\Delta_t$ with

$$\Delta_t = \frac{1}{|B_t|} \sum_{\mathbf{x}_i \in B_t} \mathbf{w}_t^i \mathbf{g}_t(\mathbf{x}_i), \mathbf{g}_t(\mathbf{x}_i) = \nabla_\theta \ell_{\text{tta}}(\mathbf{x}_i, \theta_t), \quad (1)$$

where $\mathbf{g}_t(\mathbf{x}_i)$ is the per-sample gradient, $\eta$ is the adaptation learning rate, $\ell_{\text{tta}}$ is an unsupervised/self-supervised loss (e.g., entropy minimization, consistency across transformations, or an auxiliary task), and $\mathbf{w}_t^i \in [0, 1]$ applies a data-dependent weight including filtering ($\mathbf{w}_t \in \{0, 1\}$). To prevent overfitting and improve the stability of updates, TTA often adapts only a subset of the model parameters (Wang et al., 2021; Vray et al., 2025), such as the affine parameters of normalization layers (e.g., BatchNorm (Ioffe & Szegedy, 2015)/GroupNorm (Wu & He, 2018)/LayerNorm (Ba et al., 2016)), while keeping the remaining parameters fixed. Moreover, many TTA methods explicitly regularize updates—for example by selecting/filtering samples by estimating their reliability (Niu et al., 2022; 2023)—to prevent degenerate solutions like model collapse (when a single prediction is made with high confidence on all data).

### 2.2. Differential Privacy and DP-SGD

Differential privacy (DP) (Dwork et al., 2006b) rigorously quantifies and bounds the privacy loss of individual samples through a computation, such as training a machine learning model. Intuitively, DP provides a privacy guarantee by ensuring that the output distribution of an algorithm is not sensitive to a change in any single sample. Formally:

**Definition 2.1** (($\epsilon, \delta$)-Differential Privacy (Dwork et al., 2006a))**.** A randomized algorithm $\mathcal{A}$ satisfies $(\epsilon, \delta)$-DP if, for all pairs of neighbouring dataset $(D, D')$ differing in one data point, for any measurable set of outputs $\mathcal{S}$,

$$\Pr[\mathcal{A}(D) \in \mathcal{S}] \leq e^\epsilon \Pr[\mathcal{A}(D') \in \mathcal{S}] + \delta,$$

where $\epsilon$ controls the strength of the privacy guarantee and $\delta$ allows for a small probability of failure.

A common approach to train DP deep learning models is Differentially Private Stochastic Gradient Descent (DP-SGD) (Abadi et al., 2016) that serves as a drop-in replacement for standard SGD. At each step $t$, DP-SGD clips (rescales) per-sample gradients ($\mathbf{g}_t(\mathbf{x}_i)$) to have $L2$-norm upper-bounded by a threshold $C$ ($\bar{\mathbf{g}}_t(\mathbf{x}_i)$), and adds Gaussian noise to the aggregated gradient ($\Delta_t$). That is, $\theta_{t+1} = \theta_t - \eta\Delta_t$, with:

$$\begin{aligned}
\bar{\mathbf{g}}_t(\mathbf{x}_i) &= \mathbf{g}_t(\mathbf{x}_i) / \max\left(1, \frac{||\mathbf{g}_t(\mathbf{x}_i)||_2}{C}\right) \\
\Delta_t^{\text{DP}} &= \frac{1}{|B_t|}\Big(\sum_{\mathbf{x}_i \in B_t} \bar{\mathbf{g}}_t(\mathbf{x}_i) + \mathcal{N}(0, C^2\sigma^2\mathbb{I}^d)\Big),
\end{aligned} \quad (2)$$

where the threshold $C$ limits the sensitivity of the update to a single sample $\mathbf{x}_i$; the noise multiplier $\sigma$ controls the

perturbation by scaling its variance with $C$; and the model has $d$ parameters. The overall privacy guarantee of DP-SGD is analyzed with the composition, sub-sampling, and post-processing properties of DP (Abadi et al., 2016). The cumulative privacy loss across steps is tracked by advanced accounting techniques like Gaussian Differential Privacy (GDP) (Dong et al., 2022).

While we are the first to investigate DP for updating by TTA, existing DP directions relate to updates. DP fine-tuning (Yu et al., 2021; De et al., 2022) studies private updates to pre-trained models, in a supervised setting with output labels, on a fixed dataset. DP bandits (Mishra & Thakurta, 2015) and reinforcement learning (Vietri et al., 2020) consider private updates, in a supervised setting with rewards, and methods focus on the exploration and exploitation trade-off.

### 2.3. Gradient Clipping

Gradient clipping at the batch level is a common approach to stabilize deep network optimization (Pascanu et al., 2013; Brock et al., 2021), or to handle label noise (Menon et al., 2020). While previous work (Niu et al., 2023) has found per-batch gradient clipping ineffective for TTA, we focus on per-sample clipping (Eq. 2), as a core component of differential privacy, and show that it can improve TTA methods even outside of DP (§4.3). Although we manually tune our clipping thresholds by cross-validation, as it is simple and effective, it is also possible to adaptively estimate the clipping threshold during optimization Andrew et al. (2022) for more automatic tuning.

## 3. Differentially Private Test-Time Adaptation

We first present a general recipe and analysis to make TTA updates differentially private, using Tent (Wang et al., 2021) as an example, by adopting DP gradient computation (§3.1). This approach, with customization, supports multiple popular and recent TTA methods (§3.2) that include EATA (Niu et al., 2022), SAR (Niu et al., 2023), DeYO (Lee et al., 2024) and COME (Zhang et al., 2024). The customization can involve replacing operations with DP-compatible alternatives, removing incompatible components (§3.1-§3.2), and switching to compatible deep learning architectures (§3.3). We then show that DP-TTA methods do not lose too much accuracy for their privacy, and variants can even trade privacy for better accuracy than standard TTA methods (§4).

### 3.1. Differentially Private Test-Time Updates

The key step for integrating DP into TTA is substituting the update $\Delta_t$ (Eq. 1) with the private gradient $\Delta_t^{\text{DP}}$ (Eq. 2). We first show the private method of DP-Tent (Alg. 1). Let $f_{\theta_0}$ denotes the source model (either non-private or privately-trained with DP). DP-Tent has the same entropy

---

**Algorithm 1** Test-time adaptation with DP-Tent

**Require:** Source model $f_\theta$, adaptable parameters $\theta^a$, L2-clipping threshold $C$, noise multiplier $\sigma$, learning rate $\eta$, test data $\mathbf{x}_i$ in batch $\{B_k\}_{k=1}^K$,
1: **for** $t \in \{1, \dots T\}$ **do**
2:      Calculate predictions $\hat{y}_i$ and loss $H(\mathbf{x}_i, \theta)$
3:      Compute per-example gradients $\mathbf{g}_t(\mathbf{x}_i)$ for each sample $\mathbf{x}_i$
4:      Clip each per-example gradient to get $\bar{\mathbf{g}}_t(\mathbf{x}_i)$
5:      Add noise and re-aggregate noisy clipped gradient to get $\Delta_t^{\text{DP-Tent}}$
6:      Adapt with $\theta_{t+1}^a \leftarrow \theta_t^a - \eta\, \Delta_t^{\text{DP-Tent}}$
7: **end for**
8: **return** Adapted parameters $\theta_T$

---

minimization loss as Tent $\ell_{\text{tent}}(\mathbf{x}_i, \theta)$,

$$H(\mathbf{x}_i, \theta) = -\sum_c p(\hat{y}_c) \log p(\hat{y}_c), \; \hat{y} = f_\theta(\mathbf{x}_i), \quad (3)$$

where $p(\hat{y}_c)$ is the predicted probability of class $c$. As in (Eq. 1), we optimize with a batch size $B_t$, with $\mathbf{g}_t^{\text{tent}}(\mathbf{x}_i) = \nabla_\theta \ell_{\text{tent}}(\mathbf{x}_i, \theta_t)$. DP-Tent computes a private aggregate update (with $\mathbf{w}_t^i = 1$) following (Eq. 2), clipping gradients to $\bar{\mathbf{g}}_t^{\text{tent}}(\mathbf{x}_i)$ then updating with noisy aggregate

$$\Delta_t^{\text{DP-Tent}} = \frac{1}{|B_t|} \Big( \sum_{\mathbf{x}_i \in B_t} \bar{\mathbf{g}}_t^{\text{tent}}(\mathbf{x}_i) + \mathcal{N}(0, C^2\sigma^2\mathbb{I}^d) \Big) \quad (4)$$

DP-Tent keeps the same parameterization as Tent and only updates the affine parameters of normalization layers $\theta^a \subset \theta$ as $\theta_{t+1}^a \leftarrow \theta_t^a - \eta \Delta_t^{\text{DP-Tent}}$.

**Privacy analysis.** DP-SGD typically relies on analysis based on sub-sampling due to the multi-epoch iteration and mini-batching of training optimization. TTA optimization is different, as online updates are computed on each batch only once, without resampling. Using a post-processing argument from DP, we can analyze DP-Tent's sample-level privacy with only a one step analysis. Another subtlety lies in the neighboring definition of DP (Definition 2.1). While DP-SGD for training relies on sub-sampling, and the convenient "leave-one-out" neighboring definition (DP hides the effect of any sample being included/excluded from training), TTA instead relies on a streaming sequence of test data in fixed-sized batches, where removing a sample is not as natural. In this case, the "change one" neighboring definition is a better fit: DP hides the effect of a sample being replaced by another arbitrary sample. The "change one" neighboring definition is a bit stronger, at the cost of an additional factor 2 in the DP analysis:

**Proposition 3.1** (Privacy Guarantee of DP-Tent)**.** *DP-Tent is $G_\mu$-DP with $\mu = \frac{2}{\sigma}$.*

*As a result, for all $\epsilon \geq 0$, DP-Tent is also $(\epsilon, \delta)$-DP with:*

$$\delta(\epsilon) = \Phi\Big(-\frac{\sigma\epsilon}{2} + \frac{1}{\sigma}\Big) - e^\epsilon \Phi\Big(-\frac{\sigma\epsilon}{2} - \frac{1}{\sigma}\Big),$$

*where $\Phi$ is the standard Gaussian CDF.*

*Proof.* Focusing on the sum in (Eq. 4), for neighbouring $D, D'$ that differ in $\mathbf{x}_j, \mathbf{x}'_j$, the $\ell_2$-sensitivity is $\max_{\mathbf{x}_j, \mathbf{x}'_j} \|\sum_{\mathbf{x}_i \in B_t} \bar{\mathbf{g}}_t^{\text{tent}}(\mathbf{x}_i) - \sum_{\mathbf{x}_i \in B'_t} \bar{\mathbf{g}}_t^{\text{tent}}(\mathbf{x}_i)\|_2 = \max_{\mathbf{x}_j, \mathbf{x}'_j} \|\bar{\mathbf{g}}_t^{\text{tent}}(\mathbf{x}_j) - \bar{\mathbf{g}}_t^{\text{tent}}(\mathbf{x}'_j)\|_2 \leq \max_{\mathbf{x}_j, \mathbf{x}'_j} \|\bar{\mathbf{g}}_t^{\text{tent}}(\mathbf{x}_j)\|_2 + \|\bar{\mathbf{g}}_t^{\text{tent}}(\mathbf{x}'_j)\|_2 \leq 2C$. The first equality holds because all elements in any batch are equal, except for $\mathbf{x}_j, \mathbf{x}'_j$, the first inequality by the triangular inequality, and the last inequality is enforced by per-sample gradients clipping in (Alg. 1).

By Theorem 2.7 (Dong et al., 2022), this mechanism (one step of DP-Tent) is $\mu$-GDP with $\mu = \frac{2}{\sigma}$. Since each sample is processed only once, and all further computation is post-processing on a DP result, thus remaining at least as DP, there is no privacy composition over adaptation steps.

By Corollary 2.13 of (Dong et al., 2022), DP-Tent is thus $(\epsilon, \delta)$-DP with $\delta(\epsilon) = \Phi\big(-\frac{\epsilon}{\mu} + \frac{\mu}{2}\big) - e^\epsilon \Phi\big(-\frac{\epsilon}{\mu} - \frac{\mu}{2}\big)$. Replacing the value of $\mu$ concludes the proof. $\square$

### 3.2. Integrating DP with EATA, SAR, DeYO and COME

Integrating DP with a TTA method requires making every step of its updates respect DP. Specifically, each data-dependent computation needs to be inside a DP computation (with controlled sensitivity), or done as a post-processing of a DP measurement. This way Proposition 3.1 applies and ensures DP properties. We alter multiple TTA methods to ensure DP (details in Appendix A), by replacing steps that query test data to produce intermediate results with DP versions, or removing steps when they are redundant.

**DP-EATA.** EATA makes three additions to Tent: a reweighted loss, a Fisher information regularizer $\mathcal{R}$, and filters on uncertainty and diversity to choose which samples to update on. Its Filter-1 measures uncertainty to keep samples with entropy lower than a threshold $H_0$. Its Filter-2 measures diversity by the cosine similarity of current predictions to an exponential moving average of past predictions. We use DP updates (§3.1) on the EATA loss. $\mathcal{R}$ acts only on current parameters (by DP post-processing) and original parameters (which are constant), so this addition does not impact the DP analysis. Finally, we remove both filters. Since filters can affect batch size, and require non-DP statistics, they would be costly to make DP (Appendix A confirms that DP variants preserving some of the filters have

lower accuracy). For $\lambda \geq 0$, our DP-EATA update is:

$$\mathbf{g}_t(\mathbf{x}_i) = \nabla_\theta \Big(e^{H_0 - H(\mathbf{x}_i, \theta)} H(\mathbf{x}_i, \theta_t)\Big)$$

$$\bar{\mathbf{g}}_t(\mathbf{x}_i) = \mathbf{g}_t(\mathbf{x}_i) / \max\Big(1, \frac{\|\mathbf{g}_t(\mathbf{x}_i)\|_2}{C}\Big)$$

$$\Delta_t^{\text{DP-EATA}} = \frac{1}{|B_t|}\Big(\sum_{\mathbf{x}_i \in B_t} \bar{\mathbf{g}}_t(\mathbf{x}_i) + \mathcal{N}(0, C^2 \sigma^2 \mathbb{I}^d)\Big)$$
$$+ \lambda \nabla_\theta \mathcal{R}(\theta_t, \theta_0) \tag{5}$$

$\mathcal{R}$ does not enter per-sample gradient clipping since it does not query test data, and its gradient depends only on the source parameters. Proposition 3.1 holds for DP-EATA.

**DP-SAR.** SAR also filters points based on an entropy threshold, that we also remove in DP-SAR. The sharpness minimization update in SAR involves accessing $\nabla_\theta H(\mathbf{x}_i, \theta)$ at two different points: a DP version would be costly in terms of privacy loss. We instead use a DP version from Park et al. (2023) which calculates the weight perturbation $\tilde{\epsilon}_t(\theta)$ using the last step's private gradient $\tilde{\mathbf{g}}_{t-1}$, yielding the update:

$$\mathbf{g}_t(\mathbf{x}_i) = \nabla_\theta H(\mathbf{x}_i, \theta)|_{\theta_t + \tilde{\epsilon}_t(\theta)}, \ \tilde{\epsilon}_t(\theta) = \rho\, \tilde{\mathbf{g}}_{t-1}/\|\tilde{\mathbf{g}}_{t-1}\|_2$$

$$\bar{\mathbf{g}}_t(\mathbf{x}_i) = \mathbf{g}_t(\mathbf{x}_i) / \max\Big(1, \frac{\|\mathbf{g}_t(\mathbf{x}_i)\|_2}{C}\Big)$$

$$\Delta_t^{\text{DP-SAR}} = \frac{1}{|B_t|}\Big(\sum_{\mathbf{x}_i \in B_t} \bar{\mathbf{g}}_t(\mathbf{x}_i) + \mathcal{N}(0, C^2 \sigma^2 \mathbb{I}^d)\Big)$$
$$\tag{6}$$

Proposition 3.1 holds since the per-sample contribution to the update is bounded by $C$.

**DP-DeYO.** DeYO introduces the pseudo-label probability difference (PLPD), computed on test data $\mathbf{x}_i$ and a randomly patch-shuffled version $\mathbf{x}'_i$. PLPD is used for filtering, which we remove, as we did for entropy filtering. The update also depends on the PLPD, leading to the following DP-DeYO:

$$\mathbf{g}_t(\mathbf{x}_i) = \nabla_\theta \Big((e^{H_0 - H(\mathbf{x}_i, \theta)} + e^{\text{PLPD}_\theta(\mathbf{x}_i, \mathbf{x}'_i)}) H(\mathbf{x}_i, \theta_t)\Big)$$

$$\bar{\mathbf{g}}_t(\mathbf{x}_i) = \mathbf{g}_t(\mathbf{x}_i) / \max\Big(1, \frac{\|\mathbf{g}_t(\mathbf{x}_i)\|_2}{C}\Big)$$

$$\Delta_t^{\text{DP-DeYO}} = \frac{1}{|B_t|}\Big(\sum_{\mathbf{x}_i \in B_t} \bar{\mathbf{g}}_t(\mathbf{x}_i) + \mathcal{N}(0, C^2 \sigma^2 \mathbb{I}^d)\Big)$$
$$\tag{7}$$

Since PLPD is calculated per-sample and enters the loss function through the clipping operator, the per-sample sensitivity is bounded by $C$, and Proposition 3.1 holds.

**DP-COME.** COME modifies the entropy loss with a conservative version that models uncertainty. There is no modification on the COME loss to satisfy DP:

$$\ell_{\text{COME}}(\mathbf{x}_i, \theta) = -\sum_{k=1}^{K} b_k \log b_k - u \log u,$$

$$b_k = \frac{\alpha_k - 1}{S}, \ \alpha_k = e^{f_\theta(\mathbf{x}_i)_k}, \ S = \sum_{k=1}^{K} \alpha_k, \ u = \frac{K}{S}. \tag{8}$$

The only exception is when it is used to replace $H(\mathbf{x}_i, \theta)$ in other TTA algorithms, e.g. the modifications in the DeYO algorithm to satisfy DP carry over to DP-DeYO-COME.

### 3.3. DP-Compatible Architectures

While DP does not impose general architectural constraints, the privacy analysis of DP-SGD and Proposition 3.1 relies on per-sample clipping to bound the sensitivity of updates to a change in any individual sample. This bound requires that any sample only influences its own gradient, ruling out operations which gradients depend on other samples, such as Batch Normalization (BN) (Ioffe & Szegedy, 2015). For compatibility with privacy guarantees, we exclude architectures with BN, and instead choose a common architecture with Layer Normalization (LN) (Ba et al., 2016): the Vision Transformer (ViT) (Dosovitskiy et al., 2021).

## 4. Experiments

### 4.1. Setup

We study test-time adaptation (TTA) under distribution shift, using differentially private optimizers. Unless otherwise noted, we follow the default hyperparameter choices and implementation details from each baseline method and only modify the optimizer and customizations in §3.2 for test-time updates.

**Model.** We use the ViT-Base/16 (Dosovitskiy et al., 2021) architecture with the `vit_base_patch16_224` model parameters from the `timm` library (Wightman, 2019) and ConvNeXT_Tiny (Liu et al., 2022) with the `convNeXt_Tiny` model parameters from the `timm` library. The models are pre-trained on ImageNet and then adapted at test time following the updates for each method.

**Dataset.** We evaluate on the corruptions of ImageNet-C at severity 5 (Hendrycks & Dietterich, 2019) for strong shifts and ImageNet-R (Hendrycks et al., 2021) for more shifts. To check accuracy without shift, we also evaluate on the original ImageNet validation set (Russakovsky et al., 2015). Additional results on ConvNeXt and ImageNet-R are provided in Appendix B.3.

**TTA methods.** We consider representative methods with entropy minimization and sample filtering/reweighting: Tent (Wang et al., 2021), EATA (Niu et al., 2022), SAR (Niu et al., 2023), DeYO (Lee et al., 2024), and COME (Zhang et al., 2024). Note that all adaptation updates are unsupervised, without use of test labels, and we update the same choice of affine parameters as the original methods.

**Adaptation settings.** We report results in both the episodic and continual settings. In the episodic setting, adaptation is reset and the model parameters are restored to the training parameters $\theta_0$ between each test shift. In the continual setting, adaptation proceeds over a sequence of test shifts without resets, capturing long-horizon deployment, potential error accumulation, and the difficulty of detecting shifts.

**Implementation.** All experiments are implemented with PyTorch (Paszke et al., 2019), and we use Opacus (Yousefpour et al., 2021) for per-sample gradient clipping.

**Hyperparameter tuning.** For each algorithm and experiment, we pick the best learning rate in $\{10^{-4}, 5 \cdot 10^{-4}, 10^{-3}, \ldots, 1\}$. We tune the $L2$-clipping threshold $C$ in $\{1, 5, 10, 15\}$. We fix the batch size to $64$ per the common practice following Tent. We provide additional hyperparameter sensitivity in Appendix B.5 and batch-size analyses in Appendix B.6. These results show that DP-TTA is reasonably robust to hyperparameter choices, transfers effectively when tuning on ImageNet-R, and remains stable across batch sizes, with a trade-off between DP noise and adaptation speed. For each setting, we repeat the experiment with the best hyperparameters over five random seeds and report the standard deviation in the corresponding tables. We provide a detailed variance analysis in Appendix B.4.

### 4.2. Privacy/Accuracy trade-off

We evaluate our DP-TTA algorithms (§3.1-3.2) at five noise levels $\sigma \in \{8.594, 1.966, 1.084, 0.777, 0.619\}$. For $\delta = 10^{-6}$ these correspond to $\varepsilon = 1, 5, 10, 15, 20$ privacy guarantees respectively. Smaller $\varepsilon$ indicates a stronger privacy guarantee, while $\varepsilon = \infty$ indicate non-private adaptation by the original Tent, EATA, SAR, DeYO and DeYO-COME methods.

Providing privacy guarantees often incurs a drop in model performance, as widely observed in differentially private training for deep learning applications (Abadi et al., 2016). Figure 1 shows the privacy/accuracy trade-off for DP-TTA updates in the continual adaptation setting, where the gray line in each plot shows the accuracy of the non-private method. We observe a clear privacy-accuracy trade-off across different privacy budgets, where stronger privacy guarantees (lower $\varepsilon$s) lead to lower accuracies.

Surprisingly though, DP-TTA methods can achieve comparable or even better adaptation at moderate privacy levels relative to their non-private baselines. For example, DP-Tent has an an average accuracy of $62.9\%, 62.6\%$ and $62.1\%$ at $\varepsilon = 20, 15, 10$ respectively, outperforming the non-private baseline at $60.8\%$. Even at the stronger privacy level $\varepsilon = 1$, the accuracy only degrades by $2.3\%$. This suggests that DP-Tent can provide meaningful privacy guarantees without substantially sacrificing adaptation performance. Non-private versions outperform the private versions for DP-EATA, DP-SAR, DP-DeYO and DP-DeYO-COME, though differences are only $2.9\%, 1.2\%, 2.4\%$ and $1.7\%$ respectively at $\varepsilon = 20$. For a finer view, we examine the accuracy-privacy trade-off

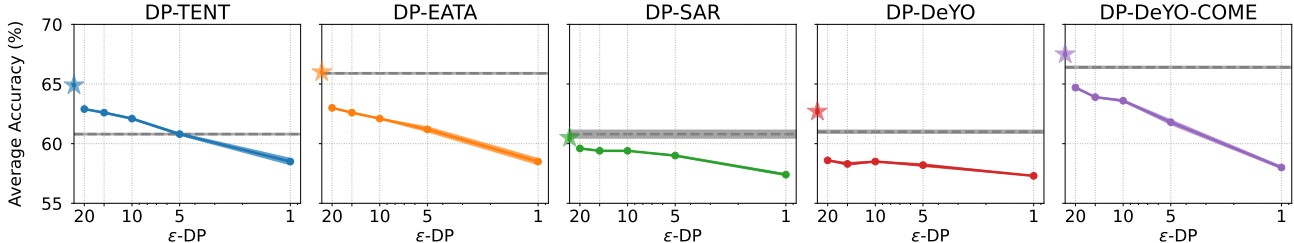

*Figure 1.* **Privacy/accuracy trade-off for DP-TTA methods at different privacy budgets in the continual setting.** The results show average top-1 accuracy (%) under different privacy budgets ($\varepsilon$-DP, $\delta = 1e^{-6}$) when adapting on the 15 corruption types of ImageNet-C at severity 5 in the continual setting. The gray dashed line and star ($\star$) are the *non-private baseline* and our *non-private + clip* editions of each method. The shaded area represent the standard deviation in the accuracy scores, where each experiment is repeated 5 times with different seed.

for each corruption type in the Appendix Sec. B.2, for the continual setting in Table 8, and for the episodic setting in Table 9, and for uncorrupted ("clean") data without shift in Table 10.

These improvements hint at a DP mechanism that could be independently useful for adaptation. A closer investigation attributes the improvement to per-sample gradient clipping, as we examine in the next section.

### 4.3. Per-sample Clipping Improves Adaptation

We apply per-sample clipping to the original editions of EATA, SAR, DeYO, and COME, preserving the sample filters, and without adding DP noise to guarantee privacy. The stars ($\star$) in Figure 1 marks the accuracy of clipping alone and show consistent improvement. Details are in Table 1.

**Main Results.** The improved accuracy (Figure 1) comes from improvements for most corruption types (Table 1) under per-sample gradient clipping, four out of five TTA methods improve, while the only exception drops marginally by 0.3%. Overall, clipping increases the average adaptation gain from 0.1% to 4.1%. DeYO-COME with clipping even achieves the highest accuracy in the continual setting with its average accuracy of 67.5%. The consistent gains due to per-sample gradient clipping suggest it may more generally improve optimization for TTA methods. The results across ImageNet-R with ViT and ImageNet-C with ConvNeXt-Tiny show trends consistent with the main results. The gains from clipping are substantial in some settings, reaching up to 14% on ImageNet-R (Table 12) and 1%–5% for Tent and EATA under ConvNeXt continual and episodic adaptation (Tables 13 and 14).

The effect of clipping is consistent with but previously undiscovered by prior work highlighting the effect of samples with high loss or large gradients (Niu et al., 2022; 2023; Zhang et al., 2024). Per-sample gradient clipping provides a general mechanism for controlling the impact of any one

sample on updates that is independent of the choice of loss or parameterization. Specifically, any large gradient with norm larger than $C$, i.e. $\|\mathbf{g}_t(x_i)\| > C$, is rescaled to have norm $C$, preserving its direction while reducing its magnitude. To analyze the effect of clipping, we compare Tent augmented with per-sample gradient clipping against existing alternatives for mitigating harmful updates in Table 2.

**Per-sample clipping has a different effect than batch-level clipping.** As mentioned in §2.3, batch-level clipping is a known method to stabilize training, which constrains the magnitude of the aggregated update by rescaling it when its norm exceeds a threshold $C$: $\theta_t \leftarrow \theta_{t-1} - \eta \operatorname{clip}(\Delta_t, C)$, with $\operatorname{clip}(\Delta_t, C) = \Delta_t / \max\left(1, \frac{\|\Delta_t\|_2}{C}\right)$.

Table 2 examines Tent with batch-level gradient clipping ("Batch-level clip"). While batch-level clipping controls the aggregated update, it does not control the impact of each gradient in the batch. A single sample with an extremely large gradient can still dominate the direction of the update, while the contributions of other samples are suppressed. In contrast, per-sample gradient clipping independently limits the contribution of each individual sample prior to aggregation, and enforces sample-wise control in the update. The type of clipping matters: batch-level clipping results in lower accuracy than other variants, including the original Tent method. The accuracy stays around $57 - 59\%$ across thresholds $C$, which range from clipping 84% of batches to just 1.5% of batches. This confirms that batch-level clipping is not effective for TTA in agreement with Niu et al. (2023).

**Clipping is more effective than excluding large gradients.** While clipping bounds the effect of large gradients by scaling, filtering removes large gradients entirely, and retains samples that satisfy $\mathbb{1}(\mathbf{g}_t(x_i) \leq C)$. Comparing "Per-sample clip" and "Filter by grad norm" in Table 2 shows that the re-scaling of per-sample gradient clipping is more effective than the hard inclusion/exclusion of filtering by gradient norm. Comparing clipping and filtering at the same threshold $C$, filtering can exclude a substantial number of

*Table 1.* **Per-sample gradient clipping improves adaptation performance across multiple TTA methods.** We add per-sample gradient clipping to each TTA method and compare with its original version and hyperparameters. The results are top-1 accuracy (%) when adapting on ImageNet-C at severity level 5 under a continual setting. The bold value signifies the top-performing result across all rows. The shaded row indicate the %improvements after adding per-sample gradient clipping to the TTA method.

| Method | Noise | | | Blur | | | | Weather | | | | Digital | | | | Avg. |
|---|---|---|---|---|---|---|---|---|---|---|---|---|---|---|---|---|
| | Gauss. | Shot | Impul. | Defoc. | Glass | Motion | Zoom | Snow | Frost | Fog | Brit. | Contr. | Elastic | Pixel | JPEG | |
| Source ($f_{\theta_0}$) | 53.9 | 53.3 | 54.1 | 49.6 | 32.3 | 52.3 | 45.4 | 60.2 | 62.4 | 65.8 | 77.3 | 36.6 | 45.2 | 67.3 | 69.3 | 55.0(0.02) |
| • Tent | 55.8 | 58.3 | 59.3 | 53.0 | 46.7 | 59.1 | 53.2 | 63.0 | 61.4 | 66.8 | 78.1 | 64.3 | 53.1 | 69.4 | 70.3 | 60.8(0.01) |
| • Tent + clip | 60.9 | 63.4 | 63.4 | 56.7 | 56.9 | 62.8 | 59.8 | 66.2 | 66.1 | 71.1 | 78.1 | 62.9 | 62.8 | 71.7 | 70.8 | 64.9 (0.05) |
| | △5.1 | △5.1 | △4.1 | △3.7 | △10.2 | △3.7 | △6.6 | △3.2 | △4.7 | △4.3 | △0.0 | △-1.4 | △9.7 | △2.3 | △0.5 | △4.1 |
| • EATA | 59.3 | 62.9 | 63.3 | 58.6 | 57.9 | 64.0 | 62.6 | 67.7 | 67.2 | 72.0 | 79.3 | 62.0 | 66.5 | 72.6 | 72.6 | 65.9 (0.07) |
| • EATA + clip | 60.3 | 63.7 | 63.6 | 57.0 | 58.2 | 63.7 | 62.6 | 67.9 | 67.1 | 72.0 | 78.4 | 64.3 | 66.8 | 73.1 | 71.5 | 66.0(0.04) |
| | △1.0 | △0.8 | △0.3 | △-1.6 | △0.3 | △-0.3 | △0.0 | △0.2 | △-0.1 | △0.0 | △-0.9 | △2.3 | △0.3 | △0.5 | △-1.1 | △0.1 |
| • SAR | 56.9 | 58.8 | 58.2 | 47.3 | 47.0 | 61.8 | 60.0 | 51.3 | 65.4 | 70.0 | 77.0 | 58.9 | 59.1 | 71.2 | 69.2 | 60.8(0.34) |
| • SAR + clip | 57.5 | 59.3 | 59.7 | 53.7 | 52.4 | 57.9 | 55.3 | 59.9 | 60.8 | 64.1 | 75.9 | 58.2 | 56.0 | 67.7 | 68.4 | 60.5(0.18) |
| | △0.6 | △0.5 | △1.5 | △6.4 | △5.4 | △-3.9 | △-4.7 | △8.6 | △-4.6 | △-5.9 | △-1.1 | △-0.7 | △-3.1 | △-3.5 | △-0.8 | △-0.3 |
| • DeYO | 57.1 | 59.0 | 59.1 | 53.6 | 52.1 | 58.2 | 54.4 | 62.4 | 61.8 | 65.5 | 76.6 | 60.1 | 58.9 | 68.1 | 68.5 | 61.0(0.12) |
| • DeYO + clip | 56.2 | 59.1 | 58.7 | 51.9 | 54.7 | 60.1 | 57.5 | 64.5 | 64.1 | 69.7 | 76.9 | 60.7 | 65.5 | 71.4 | 69.0 | 62.7(0.36) |
| | △-0.9 | △0.1 | △-0.4 | △-1.7 | △2.6 | △1.9 | △3.1 | △2.1 | △2.3 | △4.2 | △0.3 | △0.6 | △6.6 | △3.3 | △0.5 | △1.7 |
| • DeYO-COME | 61.7 | 63.7 | 63.7 | **58.7** | 58.2 | 63.9 | 60.1 | 68.3 | 67.2 | 72.6 | **79.0** | 66.3 | 67.7 | 73.4 | 71.7 | 66.4(0.10) |
| • DeYO-COME + clip | **62.0** | **64.2** | 63.8 | 58.5 | **59.1** | **65.1** | **63.9** | **70.0** | **68.6** | **74.0** | 78.9 | **66.9** | **70.3** | **74.6** | **72.1** | **67.5**(0.08) |
| | △0.3 | △0.5 | △0.1 | △-0.2 | △0.9 | △1.2 | △3.8 | △1.7 | △1.4 | △1.4 | △-0.1 | △0.6 | △2.6 | △1.2 | △0.4 | △1.1 |

*Table 2.* **Per-sample clipping has the highest average accuracy over other methods of handling difficult samples.** We report the average top-1 accuracy (and the standard deviation over 5 repeated runs) over 15 common corruption types, when adapting Imagenet-C (level 5) with ViT in the continual setting. The threshold $E_0$ used for filtering entropy loss is in the scale of $\log(10^3)$. %clip/filter indicates the percentage of samples/batch clipped or filtered, averaged over all corruption types. Bold value indicates the best result across rows.

| Method | Thre. $(C, H_0)$ | %clip/ filter | Noise | | | Blur | | | | Weather | | | | Digital | | | | Avg. |
|---|---|---|---|---|---|---|---|---|---|---|---|---|---|---|---|---|---|---|
| | | | Gauss. | Shot | Impul. | Defoc. | Glass | Motion | Zoom | Snow | Frost | Fog | Brit. | Contr. | Elastic | Pixel | JPEG | |
| Source ($f_{\theta_0}$) | / | / | 53.9 | 53.3 | 54.1 | 49.6 | 32.3 | 52.3 | 45.4 | 60.2 | 62.4 | 65.8 | 77.3 | 36.6 | 45.2 | 67.3 | 69.3 | 55.0 (0.02) |
| Tent | / | / | 52.4 | 56.3 | 58.8 | 49.6 | 51.7 | 57.4 | 53.6 | 54.5 | 60.8 | 68.6 | 77.5 | 64.9 | 52.4 | 69.2 | 68.5 | 59.7 (0.01) |
| + Per-sample clip | 1 | 100% | **60.9** | **63.4** | **63.4** | **56.7** | **56.9** | **62.8** | **59.8** | **66.2** | **66.1** | 71.1 | 78.1 | 62.9 | **62.8** | 71.7 | 70.8 | **64.9** (0.16) |
| | 5 | 35.9% | **60.9** | 63.3 | **63.4** | 55.7 | 55.6 | 62.1 | 59.2 | 65.7 | 65.4 | 70.1 | 78.2 | 62.8 | 61.5 | **72.1** | 70.8 | 64.5 (0.11) |
| | 10 | 21.0% | 60.6 | 63.0 | 62.9 | 54.7 | 54.7 | 61.7 | 57.6 | 64.9 | 65.0 | 70.4 | 78.1 | 62.7 | 60.8 | 71.8 | 70.5 | 64.0 (0.10) |
| + Batch-level clip | 1 | 84.4% | 54.9 | 57.5 | 58.6 | 51.3 | 43.2 | 58.4 | 52.2 | 62.6 | 61.3 | 65.2 | 78.0 | 61.6 | 51.7 | 69.6 | 70.3 | 59.8 (0.01) |
| | 5 | 1.7% | 55.1 | 57.5 | 58.4 | 51.8 | 42.9 | 58.4 | 54.4 | 62.0 | 60.5 | 64.0 | 77.8 | 60.5 | 50.3 | 68.4 | 69.6 | 59.2 (0.02) |
| | 10 | 1.5% | 54.4 | 55.6 | 56.3 | 51.0 | 38.3 | 56.5 | 49.8 | 60.7 | 58.7 | 63.1 | 77.5 | 56.3 | 47.7 | 67.0 | 69.3 | 57.5 (0.05) |
| + Filter by grad norm | 1 | 100% | 53.9 | 53.3 | 54.1 | 49.6 | 32.3 | 52.3 | 45.4 | 60.2 | 62.4 | 65.8 | 77.4 | 36.6 | 45.2 | 67.3 | 69.3 | 55.0 (0.003) |
| | 5 | 35.9% | 59.6 | 62.8 | 63.0 | 55.2 | 53.8 | 61.3 | 58.0 | 64.9 | 64.6 | 69.6 | 78.3 | 62.8 | 60.1 | 71.3 | 70.4 | 63.7 (0.07) |
| | 10 | 21.0% | 60.8 | 63.3 | 63.3 | 55.9 | 54.4 | 62.0 | **59.8** | 65.6 | 65.5 | 70.5 | 78.3 | 62.0 | 62.1 | 71.5 | 71.0 | 64.4 (0.18) |
| + Filter by loss | 0.05 | 69.3% | 53.9 | 53.3 | 54.1 | 49.6 | 32.3 | 52.0 | 45.1 | 64.1 | 65.9 | **71.5** | **78.9** | **65.5** | 61.5 | **72.1** | **71.8** | 59.4 (0.45) |
| | 0.1 | 40.4% | 55.3 | 59.8 | 61.2 | 55.6 | 52.4 | 61.1 | 56.9 | 64.9 | 63.1 | 69.4 | 78.4 | 65.4 | 59.0 | 70.5 | 70.9 | 62.9 (0.03) |
| | 0.3 | 8.2% | 57.7 | 60.7 | 61.8 | 54.8 | 52.7 | 60.9 | 56.0 | 63.0 | 61.7 | 69.2 | 78.1 | 64.4 | 56.0 | 70.3 | 70.0 | 62.5 (0.10) |
| + Filter by both, ∩ | 5, 0.1 | 21.6% | 58.7 | 62.6 | 63.0 | 55.5 | 54.2 | 61.5 | 58.3 | 65.1 | 65.2 | 70.6 | 78.7 | 63.8 | 60.0 | 71.7 | 70.9 | 64.0 (0.02) |
| + Filter by both, ∪ | 5, 0.1 | 41.1% | 56.5 | 62.2 | 63.1 | 55.5 | 54.9 | 61.5 | 58.6 | 65.3 | 64.8 | 70.2 | 78.7 | 62.7 | 60.7 | 71.7 | 70.7 | 63.8 (0.05) |

samples. For example, filtering with $C = 1$ excludes all samples from updates, which discard both helpful and harmful samples alike and lead to the same adaptation accuracy as the source model. In contrast, per-sample clipping with $C = 1$ continues to learn from these gradients (with reduced influence) and achieves high accuracy. This leads to improved adaptation as each sample is guaranteed to have only a bounded effect on the update while still contributing.

**Gradient norm and entropy loss filter samples differently.** Filtering the loss by only keeping samples with $\mathbb{1}(H(x;\theta) \leq H_0)$ is a popular method for limiting updates (Niu et al., 2022; 2023; Zhang et al., 2024) "Filter by Loss" in Table 2 shows the performance of Tent with this filtering. Accuracy gains are consistently smaller than with per-sample gradient clipping. Sweeping the threshold to increase the filtering rate from $8\%$ to $40\%$ yields moderate

improvements, but excluding an even more ($69\%$) does not lead to further gains. In contrast, altering a comparable proportion by per-sample clipping at $C = 1$ achieves higher accuracy without discarding samples.

While per-sample entropy loss is often correlated with per-sample gradient norm, the two quantities capture different notions of sample difficulty: higher entropy loss indicates higher prediction uncertainty and higher gradient norm indicates larger changes to the parameters. Figure 2 shows scatterplots of per-sample entropy loss versus gradient norm across 4 corruption types (chosen as those with the largest change in performance $\Delta$ in each category in Table 1). The colors indicate the temporal order of samples during continual adaptation, with lighter colors corresponding to samples seen earlier during adaptation. We observe a weak positive correlation between entropy loss and gradient norm, with no obvious pattern over time. These plots suggest that the

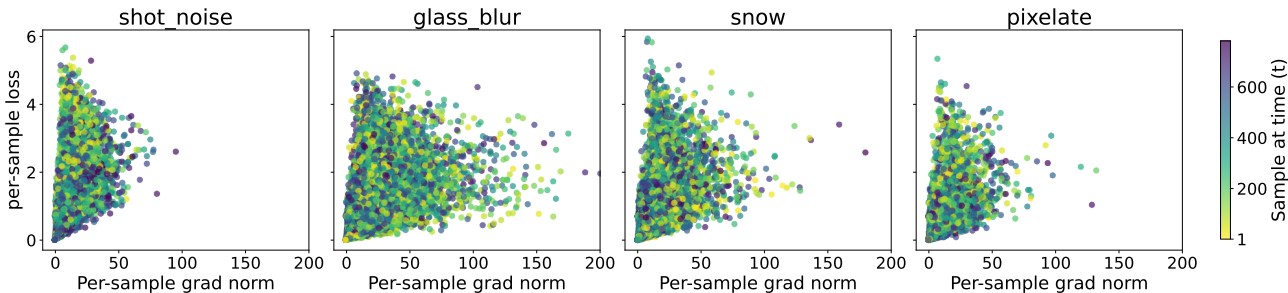

*Figure 2.* **Weak alignment between the per-sample entropy loss and gradient norm for various corruption types.** We show the scatter plot between per-sample entropy loss and gradient norm. Each point is one test sample. The colors indicate test samples received at adaptation time $t$.

*Table 3.* **Ablation study of update components** scored by top-1 accuracy (%) on ImageNet-C (severity=5). Each result is a separate configuration with equal tuning over learning rates.

| Method | Episodic | Continual |
|---|---|---|
| Source / No Update | 55.0 | 55.0 |
| Tent | 63.0 | 60.8 |
| + Clip (Ours) | 68.0 | 64.9 |
| + Regularizer | 64.5 | 62.9 |
| + Filter-1 | 66.4 | 63.1 |
| + Filter-2 | 66.5 | 65.8 |
| + Filter-1 + Filter-2 | 65.5 | 63.7 |
| + SAM | 62.5 | 60.7 |
| + SAM + Filter-1 | 63.2 | 60.9 |
| + SAM + Filter-3 | 63.0 | 60.0 |
| + SAM + Filter-1 + Filter-3 | 63.0 | 60.2 |
| + Adam | 57.4 | 62.2 |
| + Adam + Clip | 63.4 | 64.5 |

entropy loss and gradient norm identify different sets of samples to filter.

The results in Table 2 ("Filter by grad norm", "Filter by loss", "Filter by both") are consistent with the scatterplots. Filtering by both shows improves adaptation when applied as their intersection or union with each at their best threshold. The different percentages of filtered samples reveals that the filters select relatively independent sets of samples. Applying the union or intersection of both filters improves more than either filter alone. Although more effective than unfiltered updates, the hard inclusion/exclusion of the filters is less effective than per-sample gradient clipping.

**Ablations.** To analyze the role of update components, either in isolation or in combination, we compare their effect on Tent in both the episodic and continual settings in Table 3. We choose Tent for this analysis as it is the minimal method with only its simple entropy minimization loss. From EATA, we check Filter-1 that discards high-entropy samples, Filter-2 that discards samples whose updates are similar to past updates, and its regularizer for penalizing differences from

the source model. From SAR, we check its sharpness-aware minimization (SAM) update, and its Filter-3 that discards remaining high-entropy samples post-update. All hyperparameters follow the best configuration from the paper for each method, except for the learning rate, which we re-tune separately for the episodic and continual settings. This achieved the highest accuracy for each ablation tested.

For the components of EATA (Regularizer, Filter-1, Filter-2), the episodic and continual setting differ. In the episodic setting, any one component does not improve much. In the continual setting, each component helps. Its combined filters help in both settings, improving on clipping in the episodic case, and rivaling clipping in the continual case.

For the components of SAR, the SAM updates makes the larger difference, while its filters only marginally improve by ∼1 point. Nonetheless, these SAR variants do not reach the accuracy of Tent + clipping.

As an alternative to per-sample gradient clipping, an adaptive optimizer such as Adam (Kingma & Ba, 2015) re-scales updates via its estimates of gradient statistics. We check Adam without clipping, to see if its re-scaling delivers comparable improvement, and check Adam with clipping, to see if re-scaling by Adam and re-scaling by clipping are complementary. Adam does not achieve the same improvement as clipping for Tent. Furthermore, adding per-sample clipping to Adam yields a clear improvement, so re-scaling by clipping makes its own contribution beyond re-scaling by adaptive optimization. With per-sample clipping, not only the update magnitude but the update direction can change, because different samples are scaled differently.

**Computational Efficiency.** We measure the average per-batch runtime for the original methods, clipping-only variants, and differentially private (DP) variants. Clipping introduces only a small overhead (about 20–30 ms per batch) over the original methods while yielding a clear accuracy improvement. The fully DP variants requires larger implementation changes for some methods, so the runtime varies

*Table 4.* **Compute time** for non-private TTA vs. DP-TTA (average wall-clock latency per batch, ms). Measurements are taken on L40s, batch size 64, averaged over 3 runs.

| Method | Non-DP (ms) | DP (ms) | Clip-Only (ms) |
|---|---|---|---|
| Tent | 174 | 189 | 189 |
| EATA | 178 | 193 | 195 |
| SAR | 340 | 362 | 342 |
| DeYO | 246 | 287 | 264 |
| DeYO-COME | 230 | 294 | 255 |

across methods. However, the additional cost remains modest overall, while providing end-to-end privacy guarantees. Even the slowest DP variant takes only $1.28\times$ the time.

## 5. Conclusion

Private editions of test-time updates that respect DP can still reduce error, so adaptation remains possible and helpful. Non-private use of gradient clipping, specifically the per-sample clipping of DP, improves the error of multiple existing TTA methods. The small but key difference of sample-wise vs. batch-wise clipping has previously been overlooked for test-time updates, but may now find further application or extension. While there is a gap in accuracy between DP-TTA and standard TTA updates, the trade-off has now been measured, so that future work can make progress on closing it.

## Acknowledgements

We thank Francesco Croce for pre-reviewing and giving feedback. ES is supported by a Canada CIFAR AI Chair. ML is supported by the Natural Sciences and Engineering Research Council of Canada (NSERC) [reference number RGPIN-2022- 04469]. Resources used in preparing this research were provided, in part, by the Province of Ontario, the Government of Canada through CIFAR, companies sponsoring the Vector Institute, as well as the Digital Research Alliance of Canada (alliancecan.ca).

## Impact Statement

Our private test-time updates aim to produce machine learning systems that are more robust and more private. Robustness and privacy are positives, desired for trustworthy deployments of machine learning, but it is critical to note the type of robustness and privacy provided. The robustness of these test-time updates is empirical: it is measured by experiment but not guaranteed, as is the case for existing test-time adaptation methods. The privacy guarantee of these updates is theoretical: it is backed by differential privacy, but only guarantees this type of privacy. We evaluate on standard benchmarks for visual recognition and adaptation, and so we do not influence the choice of tasks for better or worse. By studying the intersection of test-time adaptation and differential privacy this work encourages further progress on robustness and privacy.

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

# A. More Details on Integrating DP with EATA, SAR, DeYO and COME

This section presents more details on the EATA (Niu et al., 2022), SAR (Niu et al., 2023), DeYO (Lee et al., 2024) and COME (Zhang et al., 2024) algorithms, and their corresponding DP variants.

**EATA** extends the entropy minimization objective in Tent with two sample selection criteria and adds regularization to reduce error accumulation and catastrophic forgetting:

$$\Delta_t^{\text{EATA}} = \frac{1}{|B_t|} \sum_{\mathbf{x}_i \in B_t} \nabla_\theta(\mathbf{w}_i H(\mathbf{x}_i, \theta)) + \lambda \nabla_\theta \mathcal{R}(\theta_t, \theta_0)$$

$$\mathbf{w}_i = e^{(H_0 - H(\mathbf{x}_i, \theta))} \mathbb{1}_{\{H(\mathbf{x}_i, \theta) < H_0\}}(\mathbf{x}_i) \cdot \mathbb{1}_{\{\cos(f_\theta(\mathbf{x}_i), m^{t-1}) < \epsilon\}}(\mathbf{x}_i)$$

$$\mathcal{R}(\theta_t, \theta_0) = \sum_{\theta_i \in \theta} \omega(\theta_{i,0})(\theta_i - \theta_{i,0}), \ \omega(\theta_{i,0}) = \frac{1}{Q} \sum_{x_q \in D_F} (\frac{\partial}{\partial \theta_0} \mathcal{L}_{CE}(f_{\theta_0}(x_q), \hat{y}_q))^2$$

The two indicator functions $\mathbb{1}(x_i)$ denotes the two sample filters: $\mathbb{1}_{\{H(\mathbf{x}_i, \theta) < H_0\}}(x_i)$ select samples with confident predictions when the per-sample entropy loss $H(\mathbf{x}_i, \theta)$ is smaller than a pre-defined threshold $H_0$; $\mathbb{1}_{\cos(f_\theta(\mathbf{x}_i), m^{t-1}) < \epsilon}(x)$ selects samples with diverse model outputs based on the cosine similarity between the current prediction to a moving average $m^t = \alpha(1/n \sum_{i=1}^n \hat{y}_i^t) + (1 - \alpha)m^{t-1}$. Samples that are filtered do not enter the calculation of mean entropy loss thus do not participate in adapting model parameters. $\mathcal{R}(\theta_t, \theta_0)$ is a Fisher regularizer where $\theta_0$ denotes the initial model parameter from the source model. $D_F = \{x_q\}_{q=1}^Q$ is a subset of test data of the source model that is not overlapping with the test data to adapt, and $\hat{y}_q$ is there corresponding predictions from the source model. The weights $\omega(\theta_i^o)$ are computed once before adaptation begins and fixed thereafter. $\lambda$ is a non-negative constant for adjusting the strength of regularization.

Both sample filters violate DP as they require computing statistics based on test data $x_i$, e.g. $H(\mathbf{x}_i, \theta)$, $f_\theta(\mathbf{x}_i)$ and $m^{t-1}$, and changes the effective batch size $B_t$. We propose to remove these two steps for **DP-EATA**. The source model parameters $\theta_0$ and data $D_F$ are considered as public information, thus the regularizer $\mathcal{R}$ does not incur privacy loss. The update of DP-EATA is:

$$\mathbf{g}_t(\mathbf{x}_i) = \nabla_\theta \left( e^{H_0 - H(\mathbf{x}_i, \theta)} H(\mathbf{x}_i, \theta_t) \right), \ \bar{\mathbf{g}}_t(\mathbf{x}_i) = \mathbf{g}_t(\mathbf{x}_i) / \max \left( 1, \frac{\|\mathbf{g}_t(\mathbf{x}_i)\|_2}{C} \right)$$

$$\Delta_t^{\text{DP-EATA}} = \frac{1}{|B_t|} \left( \sum_{\mathbf{x}_i \in B_t} \bar{\mathbf{g}}_t(\mathbf{x}_i) + \mathcal{N}(0, C^2 \sigma^2 \mathbb{I}^d) \right) + \lambda \nabla_\theta \mathcal{R}(\theta_t, \theta_0)$$

The privacy analysis follows from Proposition 3.1.

**SAR** uses the same sample filter as the first filter in EATA, and further improves robustness by incorporating sharpness aware optimization to the target of entropy minimization, that finds flat minima for better generalization. The update of SAR is:

$$\mathcal{L}^{\text{SAR}} = \frac{1}{|B_t|} \sum_{\mathbf{x}_i \in B_t} \mathbb{1}_{\{H(\mathbf{x}_i, \theta) < H_0\}}(\mathbf{x}_i) \ell_{\text{sar}}(\mathbf{x}_i, \theta), \ \ell_{\text{sar}}(\theta, \mathbf{x}_i) = \max_{\|\epsilon\|_2 \leq \rho} H(\mathbf{x}_i, \theta + \epsilon).$$

The sharpness minimization optimization is performed as follows. It first calculates the optimal weight perturbation given the gradient $\nabla_\theta H(\mathbf{x}_i, \theta)$, by solving a dual norm problem (Foret et al., 2020). The solution is given by $\epsilon(\theta) = \rho \, \text{sign}(\nabla_\theta H(x, \theta)) |\nabla_\theta H(x, \theta)| / \|\nabla_\theta H(x, \theta)\|_2$. The model parameters are then updated with:

$$\Delta_t^{\text{SAR}} = \frac{1}{|B_t|} \sum_{\mathbf{x}_i \in B_t} \mathbb{1}_{\{H(\mathbf{x}_i, \theta) < H_0\}}(\mathbf{x}_i) \nabla_\theta H(\mathbf{x}_i, \theta)|_{\theta + \epsilon(\theta)}$$

For the same reason as explained in DP-EATA, we first remove the sample filtering criteria $\mathbb{1}_{\{H(\mathbf{x}_i, \theta) < H_0\}}(\mathbf{x}_i)$ in **DP-SAR**. As the sharpness minimization problem involves accessing $H(\mathbf{x}_i, \theta)$ and its gradient, it need to be privatized to ensure an end-to-end privacy guarantee. We substitute these steps following a differentially private version of sharpness-aware learning (DP-SAT, Park et al. (2023)), which the weight perturbation $\tilde{\epsilon}_t(\theta)$ is calculated using the last step's private gradient, and $\tilde{\mathbf{g}}_{t-1}$. The resulting DP-SAR update is:

$$\mathbf{g}_t(\mathbf{x}_i) = \nabla_\theta H(\mathbf{x}_i, \theta_t)|_{\theta + \tilde{\epsilon}_t(\theta)}, \ \tilde{\epsilon}_t(\theta) = \rho \tilde{\mathbf{g}}_{t-1} / \|\tilde{\mathbf{g}}_{t-1}\|_2, \ \bar{\mathbf{g}}_t(\mathbf{x}_i) = \mathbf{g}_t(\mathbf{x}_i) / \max \left( 1, \frac{\|\mathbf{g}_t(\mathbf{x}_i)\|_2}{C} \right)$$

$$\Delta_t^{\text{DP-SAR}} = \frac{1}{|B_t|} \left( \sum_{\mathbf{x}_i \in B_t} \bar{\mathbf{g}}_t(\mathbf{x}_i) + \mathcal{N}(0, C^2 \sigma^2 \mathbb{I}^d) \right)$$

Finally, SAR recovers model parameters to the initial state $\theta_0$ by accumulating a moving average of entropy losses and comparing it to a pre-defined threshold, i.e. SAR resets $\theta_t$ to $\theta_0$ when $e_m < e_0$, where $e_m = 0.9 \times e_m + 0.1 \times H(\mathbf{x}_i, \theta)$. Such step prevents model collapse under a few extremely hard samples. The moving average of clean predictions $H(\mathbf{x}_i, \theta)$ violates the DP guarantee in Proposition 3.1. We propose to remove such model recovery step as per-sample gradient clipping provides a similar control on preventing hard samples from collapsing the model. Empirical performances in Table 1 and 6 provides evidence that DP-SAR has better adaptation performances without recovering model parameters.

**DeYO** enhances the sample filters and sample weighting strategies with an additional criterion based on pseudo-label probability difference (PLPD):

$$\Delta_t^{\text{DeYO}} = \frac{1}{|B_t|} \sum_{\mathbf{x}_i \in B_t} \nabla_\theta \big( (e^{H_0 - H(\mathbf{x}_i, \theta)} + e^{\text{PLPD}_\theta(\mathbf{x}_i, \mathbf{x}_i')}) \cdot \mathbb{1}_{\{H(\mathbf{x}_i, \theta) < H_0, \, \text{PLPD}_\theta(\mathbf{x}_i, \mathbf{x}_i') > \tau\}}(\mathbf{x}_i) H(\mathbf{x}_i, \theta) \big)$$

$$\text{PLPD}_\theta(\mathbf{x}_i, \mathbf{x}_i') = (f_\theta(\mathbf{x}_i) - f_\theta(\mathbf{x}_i'))_{\hat{y}}, \, \hat{y} = \arg\max f_\theta(\mathbf{x}_i)$$

$x$ denotes the original test data, and $x'$ denotes a randomly patch-shuffled version of $x$. $\tau$ is a pre-defined threshold for PLPD. $\hat{y}$ is a pseudo-label estimated by the prediction of $\mathbf{x}_i$ from current model $f$.

We first similarly remove the sample filter $\mathbb{1}_{\{H(\mathbf{x}_i, \theta) < H_0, \, \text{PLPD}_\theta(\mathbf{x}_i, \mathbf{x}_i') > \tau\}}(\mathbf{x}_i)$ for **DP-DeYO**. Since $\mathbf{x}_i'$ is a randomly shuffled image of $\mathbf{x}_i$, and $\text{PLPD}_\theta(\mathbf{x}_i, \mathbf{x}_i')$ is computed per-sample pair $(\mathbf{x}_i, \mathbf{x}_i')$, its sensitivity can be bounded when performing per-sample gradient clipping on the weighted loss. The loss function for DP-DeYO is then

$$\mathbf{g}_t(\mathbf{x}_i) = \nabla_\theta \Big( (e^{H_0 - H(\mathbf{x}_i, \theta)} + e^{\text{PLPD}_\theta(\mathbf{x}_i, \mathbf{x}_i')}) H(\mathbf{x}_i, \theta_t) \Big), \, \bar{\mathbf{g}}_t(\mathbf{x}_i) = \mathbf{g}_t(\mathbf{x}_i) / \max \Big( 1, \frac{\|\mathbf{g}_t(\mathbf{x}_i)\|_2}{C} \Big)$$

$$\Delta_t^{\text{DP-DeYO}} = \frac{1}{|B_t|} \Big( \sum_{\mathbf{x}_i \in B_t} \bar{\mathbf{g}}_t(\mathbf{x}_i) + \mathcal{N}(0, C^2 \sigma^2 \mathbb{I}^d) \Big)$$

**COME** proposes to modify the entropy loss used in TTA methods with a conservative version that models uncertainty, i.e. $\min_\theta H(\mathbf{x}_i, \theta)$ subject to $|u_\theta(\mathbf{x}_i) - u_{\theta_0}(\mathbf{x}_i)| \leq \delta$, where $u$ is the uncertainty estimate, and $\delta$ is a threshold. Specifically, COME proposes the *entropy of opinion* loss as:

$$\mathcal{L}_{\text{COME}} = \frac{1}{|B_t|} \sum_{\mathbf{x}_i \in B_t} \Big( -\sum_{k=1}^{K} b_k \log b_k - u \log u \Big)$$

$$b_k = \frac{\alpha_k - 1}{S}, \, \alpha_k = e^{f(x)_k}, \, S = \sum_{k=1}^{K} \alpha_k, \, u = \frac{K}{S}$$

To constrain the uncertainty mass, COME shows that constraining $u$ can be done by constraining the norm of the model output logits (Lemma 1 Zhang et al. (2024)). Let $f(x)$ be the logits of $x$, $k \in \{1, \ldots K\}$ denotes the class index, COME reparameterizes the logit and adds a stop-gradient on the logit norm:

$$f(x) \leftarrow \frac{f(x)}{\|f(x)\|_2} \cdot \|f(x)\|_2^{\text{no grad}}$$

Such logit then enters the $\alpha_k$ term in calculating the COME loss. There is no modification on the COME loss to satisfy DP except when it is used to replace $H(\mathbf{x}_i, \theta)$ in other TTA algorithms, e.g. the modifications in the DeYO algorithm to satisfy DP carry over to DP-DeYO-COME.

**Alternative approach of integrating DP with a specific TTA algorithm.** An alternative approach to removing sample filters while preserving privacy guarantees is to fix the effective batch size $B_t$ at a pre-defined constant batch size $B$. This applies to sample filters that computes criterion at per-sample level, e.g. $\mathbb{1}_{\{H(\mathbf{x}_i, \theta) < H_0\}}(\mathbf{x}_i)$, and $\mathbb{1}_{\text{PLPD}_\theta(\mathbf{x}_i, \mathbf{x}_i') > \tau\}}(\mathbf{x}_i)$.

Using entropy-loss sample filter as an example, selecting samples with $H(\mathbf{x}_i, \theta) \geq H_0$ reduces the size of $|B_t|$ when computing the mean loss over a batch. Since $|B_t|$ (and thus the mean loss value) depends on the collective inclusion or exclusion decisions across all samples in the batch, such filtering violates bounded per-sample sensitivity. In contrast, when $B_t$ is fixed at $B$, the loss computation depends on each sample independently, where the contribution of any individual sample can be bounded through per-sample gradient clipping with $L2$-threshold $C$. Therefore, under such fixed-denominator

formulation, the resulting update remains compatible with DP. We evaluate this variant of DP-EATA-V2, DP-SAR-V2 and DP-DeYO-V2 in Table 5. Comparing to results in Table 8, we observe no significant difference or worse performance from these $V2$ variants.

Sample filters that rely on moving averages or statistics aggregated across multiple samples, e.g. $\mathbb{1}_{\{\cos(f_\theta(\mathbf{x}_i), m^{t-1}) < \epsilon\}}(\mathbf{x}_i)$ in EATA, are generally more challenging to preserving its intended effect while making it DP-compatible. We leave the investigation of effective integration of DP with such algorithm-specific filtering mechanisms to future work.

*Table 5.* **Alternative design of DP-TTA algorithms.** The DP-EATA-V2, DP-SAR-V2 and DP-DeYO-V2 are variants with sample filtering steps included and a fixed batch size. The results are top-1 accuracy (%) when adapting on ImageNet-C at severity level 5 under a continual setting. We observe no significant difference or worse performance from these V2 variants.

| Method | $\varepsilon$-DP | Noise | | | Blur | | | | Weather | | | | Digital | | | | Avg. |
| | | Gauss. | Shot | Impul. | Defoc. | Glass | Motion | Zoom | Snow | Frost | Fog | Brit. | Contr. | Elastic | Pixel | JPEG | |
|---|---|---|---|---|---|---|---|---|---|---|---|---|---|---|---|---|---|
| DP-EATA-V2 | 20 | 58.2 | 59.7 | 60.0 | 57.3 | 56.5 | 62.9 | 61.0 | 68.1 | 66.0 | 66.6 | 78.5 | 61.3 | 68.6 | 74.0 | 72.0 | 64.7 |
| | 15 | 56.8 | 58.3 | 58.7 | 55.1 | 55.1 | 61.1 | 59.5 | 66.9 | 64.7 | 64.5 | 77.8 | 58.3 | 67.5 | 73.2 | 71.1 | 63.2 |
| | 10 | 58.1 | 59.3 | 59.7 | 56.6 | 51.3 | 61.0 | 57.2 | 65.8 | 63.6 | 65.7 | 78.3 | 59.4 | 63.1 | 72.2 | 71.3 | 62.8 |
| | 5 | 57.0 | 58.3 | 58.7 | 54.7 | 50.5 | 59.6 | 55.7 | 65.1 | 62.4 | 59.6 | 77.9 | 56.4 | 62.3 | 71.7 | 70.8 | 61.4 |
| | 1 | 54.7 | 55.6 | 55.9 | 51.6 | 37.8 | 55.8 | 48.8 | 61.3 | 57.4 | 66.6 | 77.4 | 43.3 | 48.7 | 68.4 | 69.5 | 56.9 |
| DP-SAR-V2 | 20 | 56.4 | 58.4 | 58.7 | 52.2 | 47.8 | 56.3 | 51.7 | 61.3 | 61.0 | 62.5 | 76.4 | 57.1 | 55.0 | 66.4 | 68.2 | 59.3 |
| | 15 | 54.8 | 56.9 | 57.9 | 51.2 | 43.0 | 57.3 | 50.4 | 61.7 | 60.8 | 65.4 | 77.8 | 59.3 | 52.3 | 67.5 | 69.4 | 59.0 |
| | 10 | 54.7 | 56.9 | 57.8 | 51.2 | 42.9 | 57.3 | 50.4 | 61.7 | 60.8 | 65.3 | 77.8 | 59.2 | 52.4 | 67.4 | 69.4 | 59.0 |
| | 5 | 54.6 | 56.7 | 57.7 | 51.4 | 42.5 | 57.2 | 50.2 | 61.4 | 60.7 | 64.8 | 77.7 | 58.0 | 52.3 | 67.1 | 69.2 | 58.8 |
| | 1 | 53.9 | 54.5 | 55.0 | 50.1 | 36.0 | 55.0 | 48.5 | 60.6 | 60.4 | 66.3 | 77.5 | 51.5 | 48.1 | 67.3 | 69.4 | 56.9 |
| DP-DeYO-V2 | 20 | 54.9 | 56.8 | 57.5 | 50.7 | 42.4 | 56.6 | 49.6 | 60.8 | 61.1 | 65.0 | 77.4 | 58.0 | 52.1 | 65.8 | 68.9 | 58.5 |
| | 15 | 54.9 | 56.8 | 57.5 | 50.7 | 42.4 | 56.5 | 49.6 | 60.8 | 61.1 | 65.0 | 77.4 | 57.9 | 52.1 | 65.9 | 69.0 | 58.5 |
| | 10 | 54.9 | 56.8 | 57.5 | 50.7 | 42.4 | 56.5 | 49.7 | 60.8 | 61.0 | 64.9 | 77.4 | 57.7 | 52.1 | 65.9 | 69.0 | 58.5 |
| | 5 | 54.9 | 56.8 | 57.6 | 50.8 | 42.3 | 56.5 | 49.7 | 60.6 | 60.9 | 64.6 | 77.4 | 57.3 | 51.8 | 66.0 | 68.9 | 58.4 |
| | 1 | 54.4 | 56.0 | 56.9 | 51.0 | 40.0 | 56.1 | 49.3 | 59.7 | 59.8 | 63.2 | 77.5 | 54.8 | 49.6 | 67.3 | 69.2 | 57.7 |

# B. More Experiments and Results

## B.1. TTA with Per-sample Gradient Clipping in Episodic Setting

Table 6 demonstrates that adding per-sample gradient clipping to the TTA methods have similarly improved performances in the episodic adaptation setting (where the model parameter is reset to $\theta_0$ when switching to a different corruption type). We observe that DeYO-COME + clip has the best performance overall, with a $69.1\%$ average accuracy over all 15 corruption types. We observe a significant improvement in average accuracy overall, with the highest of $5\%$ when adding per-sample gradient clipping to Tent. Table 7 further ablates the effect of per-sample clipping when combining with Tent, to other alternative methods that control hard sample influence to the model updates. Similar to the continual adaptation case, we observe that Tent + clip has the best accuracy for most corruption types. Such results indicate that per-sample gradient clipping can be an effective add-on to existing TTA methods for improved adaptation performances.

## B.2. Privacy-Accuracy Trade-off by Corruption Type

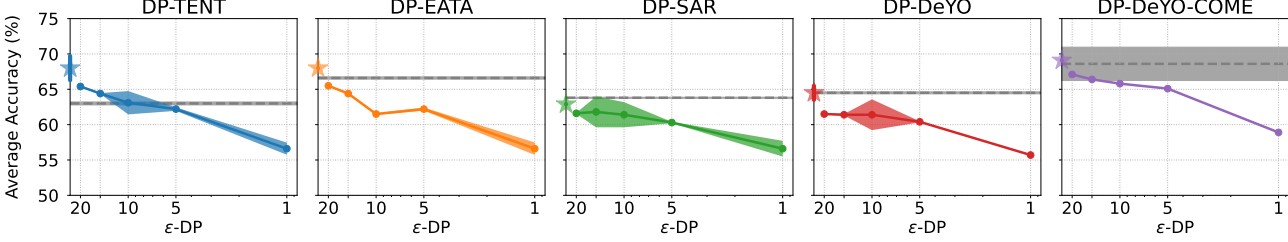

*Figure 3.* **Privacy/accuracy trade-off for DP-TTA methods at different privacy budgets in the episodic setting.** The results show average top-1 accuracy (%) under different privacy budgets ($\varepsilon$-DP, $\delta = 1e^{-6}$) when adapting on the 15 corruption types of ImageNet-C at severity 5 in the continual setting. The gray dashed line and star ($\star$) are the *non-private baseline* and our *non-private + clip* editions of each method. The shaded area represent the standard deviation in the accuracy scores, where each experiment is repeated 5 times with different seed.

*Table 6.* **Per-sample gradient clipping improves adaptation performance across multiple TTA methods.** We add per-sample gradient clipping to each TTA method and compare with its original version. The results are top-1 accuracy (%) when adapting on ImageNet-C at severity level 5 under an **episodic** setting. The bold value signifies the top-performing result across all rows. The shaded row indicate the %improvements after adding per-sample gradient clipping to the TTA method.

| Method | Noise | | | Blur | | | | Weather | | | | Digital | | | | Avg. |
|---|---|---|---|---|---|---|---|---|---|---|---|---|---|---|---|---|
| | Gauss. | Shot | Impul. | Defoc. | Glass | Motion | Zoom | Snow | Frost | Fog | Brit. | Contr. | Elastic | Pixel | JPEG | |
| Source ($f_{\theta_0}$) | 53.9 | 53.3 | 54.1 | 49.6 | 32.3 | 52.3 | 45.4 | 60.2 | 62.4 | 65.8 | 77.3 | 36.6 | 45.2 | 67.3 | 69.3 | 55.0(0.02) |
| • Tent | 58.7 | 59.7 | 60.1 | 58.8 | 53.8 | 62.6 | 58.7 | 54.6 | 59.7 | 70.0 | 78.6 | 66.1 | 60.3 | 72.2 | 71.2 | 63.0(0.21) |
| • Tent + clip | 60.8 | 62.5 | 62.2 | 61.1 | 60.0 | 66.7 | 64.9 | 71.0 | 68.8 | 73.8 | 79.8 | 67.2 | 71.8 | 76.2 | 73.5 | 68.0(1.72) |
| | △2.1 | △2.8 | △2.1 | △2.3 | △6.2 | △4.1 | △6.2 | △16.4 | △9.1 | △3.8 | △1.2 | △1.1 | △11.5 | △4.0 | △2.3 | △5.0 |
| • EATA | 60.3 | 61.5 | 61.7 | 59.9 | 58.3 | 65.4 | 63.4 | 69.1 | 67.1 | 72.1 | 79.3 | 64.3 | 69.3 | 74.7 | 72.7 | 66.6 (0.16) |
| • EATA + clip | 61.0 | 62.5 | 62.4 | 61.1 | 60.1 | 66.6 | 65.4 | 71.2 | 69.0 | 73.5 | 79.8 | 66.1 | 72.1 | 76.2 | 73.7 | 68.0(0.03) |
| | △0.7 | △1.0 | △0.7 | △1.2 | △1.8 | △1.2 | △2.0 | △2.1 | △1.9 | △1.4 | △0.5 | △1.8 | △2.8 | △1.5 | △1.0 | △1.4 |
| • SAR | 57.5 | 59.4 | 59.5 | 57.6 | 56.5 | 63.2 | 61.5 | 57.0 | 64.3 | 67.0 | 77.6 | 63.6 | 67.6 | 73.5 | 71.5 | 63.8 (0.08) |
| • SAR + clip | 57.5 | 58.6 | 58.8 | 57.2 | 53.0 | 61.0 | 57.5 | 64.7 | 62.3 | 67.4 | 78.1 | 62.4 | 61.9 | 71.5 | 70.8 | 62.9(0.02) |
| | △0.0 | △-0.8 | △-0.7 | △-0.4 | △-3.5 | △-2.2 | △-4.0 | △7.7 | △-2.0 | △0.4 | △0.5 | △-1.2 | △-5.7 | △-2.0 | △-0.7 | △-0.9 |
| • DeYO | 56.8 | 58.8 | 58.8 | 57.1 | 56.4 | 63.7 | 61.2 | 68.0 | 63.8 | 68.3 | 78.3 | 63.1 | 68.6 | 73.5 | 71.4 | 64.5(0.14) |
| • DeYO + clip | 56.7 | 58.8 | 57.9 | 56.1 | 55.4 | 62.8 | 60.8 | 67.9 | 65.2 | 70.9 | 77.7 | 62.5 | 69.5 | 73.6 | 70.9 | 64.5(1.00) |
| | △-0.1 | △0.0 | △-0.9 | △-1.0 | △-1.0 | △-0.9 | △-0.4 | △-0.1 | △1.4 | △2.6 | △-0.6 | △-0.6 | △0.9 | △0.1 | △-0.5 | △0.0 |
| • DeYO-COME | 61.9 | 63.2 | 63.0 | 61.6 | 60.7 | 67.1 | 65.6 | 71.8 | 69.0 | 74.2 | **80.1** | 68.1 | 71.9 | **76.5** | **73.9** | 68.6(2.36) |
| • DeYO-COME + clip | **62.0** | **63.6** | **63.2** | **62.4** | **61.5** | **67.9** | **66.8** | **72.5** | **70.1** | **75.0** | 80.0 | **68.4** | **73.2** | **76.5** | 73.8 | **69.1**(0.04) |
| | △0.1 | △0.4 | △0.2 | △0.8 | △0.8 | △0.8 | △1.2 | △0.7 | △1.1 | △0.8 | △-0.1 | △0.3 | △1.3 | △0.0 | △-0.1 | △0.5 |

*Table 7.* **Per-sample clipping has better accuracy over other methods of handling difficult samples.** We report the average top-1 accuracy (and the standard deviation over 5 repeated runs) over 15 common corruption types, when adapting Imagenet-C (level 5) with ViT in the **episodic** setting. The threshold $E_0$ used for filtering entropy loss is in the scale of $\log(10^3)$. %clip/filter indicates the percentage of samples/batch clipped or filtered, averaged over all corruption types. Bold value indicates the best result across rows.

| Method | Thre. $(C, H_0)$ | %clip/ filter | Noise | | | Blur | | | | Weather | | | | Digital | | | | Avg. |
|---|---|---|---|---|---|---|---|---|---|---|---|---|---|---|---|---|---|---|
| | | | Gauss. | Shot | Impul. | Defoc. | Glass | Motion | Zoom | Snow | Frost | Fog | Brit. | Contr. | Elastic | Pixel | JPEG | |
| Source ($f_{\theta_0}$) | / | / | 53.9 | 53.3 | 54.1 | 49.6 | 32.3 | 52.3 | 45.4 | 60.2 | 62.4 | 65.8 | 77.3 | 36.6 | 45.2 | 67.3 | 69.3 | 55.0 (0.02) |
| Tent | / | / | 58.7 | 59.7 | 60.1 | 58.8 | 53.8 | 62.6 | 58.7 | 54.6 | 59.7 | 70.0 | 78.6 | 66.1 | 60.3 | 72.2 | 71.2 | 63.0(0.21) |
| + Per-sample clip | 1 | 100% | 60.7 | **62.4** | **62.1** | **61.2** | **60.2** | **66.6** | **65.2** | **71.1** | **68.8** | **73.9** | 79.8 | **67.0** | 71.9 | **76.2** | **73.8** | **68.1**(0.02) |
| | 5 | 49.3% | **60.9** | 61.9 | **62.1** | 61.0 | 58.5 | 65.6 | 63.4 | 69.5 | 66.9 | 73.1 | **79.9** | **67.0** | 69.5 | 75.5 | 73.3 | 67.2(0.02) |
| | 10 | 15.7% | 60.4 | 62.0 | 61.9 | 60.7 | 59.2 | 66.0 | 63.8 | 20.0 | 11.4 | 73.6 | 80.0 | 66.9 | 70.3 | 76.0 | 73.5 | 60.4 (1.56) |
| + Batch-level clip | 1 | 81.8% | 57.7 | 58.9 | 58.5 | 55.6 | 49.4 | 60.3 | 55.3 | 57.7 | 60.0 | 66.0 | 78.4 | 59.6 | 57.2 | 71.9 | 70.9 | 61.2 (1.70) |
| | 5 | 2.5% | 55.6 | 56.6 | 56.8 | 53.5 | 42.3 | 57.3 | 51.2 | 61.1 | 56.5 | 60.2 | 77.6 | 57.3 | 49.3 | 69.0 | 69.7 | 58.3 (1.49) |
| | 10 | 0.0% | 55.0 | 56.4 | 55.9 | 52.0 | 41.5 | 56.5 | 50.2 | 60.5 | 55.9 | 59.3 | 77.4 | 55.6 | 48.7 | 68.9 | 69.4 | 57.5 (1.33) |
| + Filter by loss | 0.1 | 77.6% | 54.9 | 56.1 | 55.7 | 53.9 | 43.0 | 57.1 | 50.4 | 61.7 | 63.0 | 65.8 | 77.5 | 36.6 | 50.6 | 68.6 | 69.7 | 57.6 (0.09) |
| | 0.3 | 8.5% | 60.5 | 61.7 | 61.9 | 60.3 | 58.5 | 65.3 | 63.2 | 69.2 | 67.3 | 71.7 | 79.6 | 65.0 | 68.7 | 75.0 | 72.9 | 66.7 (2.87) |
| | 0.6 | 2.7% | 57.9 | 58.8 | 59.1 | 57.6 | 51.1 | 61.1 | 56.6 | 62.6 | 60.7 | 65.9 | 78.3 | 65.2 | 57.1 | 71.1 | 70.5 | 62.2 (0.11) |
| + Filter by grad norm | 1 | 100% | 53.9 | 53.3 | 54.1 | 49.6 | 32.3 | 52.3 | 45.4 | 60.2 | 62.4 | 65.8 | 77.4 | 36.6 | 45.2 | 67.3 | 69.3 | 55.0 (0.01) |
| | 5 | 49.3% | 59.5 | 61.0 | 60.6 | 57.7 | 52.5 | 62.6 | 59.4 | 67.8 | 64.3 | 65.5 | 79.1 | 52.5 | 66.2 | 73.7 | 72.4 | 63.7 (0.18) |
| | 10 | 15.7% | 59.3 | 60.9 | 60.3 | 57.8 | 57.9 | 64.4 | 63.3 | 70.2 | 68.0 | 72.3 | 79.3 | 0.5 | **72.2** | 75.6 | 73.0 | 62.4 (0.14) |
| + Filter by both, ∩ | 5,0.3 | 4.3% | 59.7 | 61.7 | 61.1 | 59.4 | 59.1 | 65.7 | 64.8 | 71.0 | **68.8** | 73.3 | 79.7 | 64.5 | 71.8 | 76.0 | 73.6 | 67.3 (0.25) |
| + Filter by both, ∪ | 5,0.3 | 42.8% | 60.1 | 62.1 | 61.5 | 59.7 | 58.5 | 65.1 | 63.4 | 70.2 | 67.8 | 65.8 | 79.6 | 24.6 | 71.0 | 75.6 | 73.1 | 63.9 (0.63) |

Table 8 and 9 shows the detailed results of the privacy-utility tradeoff decomposed with per-corruption-type accuracies, under the continual and episodic adaptation settings respectively. We add results from two non-private baselines with $\varepsilon = \infty$, the TTA method itself and TTA + clip, for easier comparisons. In general, the privacy-utility tradeoff applies: for most corruption types under both adaptation settings, we observe a decreasing average accuracy with stronger privacy guarantee (lower $\varepsilon$). However, due to the strong advantage from per-sample gradient clipping alone, DP-Tent can achieve comparable or sometimes better performance than the non-private (not-clipped) baselines.

To ensure that our method does not degrade performance on clean data, we evaluate it on the original (clean) test set using the hyperparameters that yield the best DP-utility accuracy in the continual setting and compare against the original baseline. The results are reported in Table 10. Under the two most stringent privacy budgets, the drop in clean accuracy is small.

## B.3. Results on the ImageNet-R Dataset and ConvNeXt Model

To check the generality of our results we perform the same main experiments on a new dataset, ImageNet-R, and a new model from a different family, ConvNeXt. Table 12 shows the accuracy of the baseline adaptation method and its clipping and DP variants at different privacy levels when adapting the ImageNet-R dataset with ViT source model. The results are similar to those on ImageNet-C, where DP-Tent outperform non-private (not-clipped) Tent baseline at the strongest $\epsilon = 1$ privacy level, and DP-EATA, DP-SAR, DP-DeYO, DP-DeYO-COME outperform the baseline at lower privacy

*Table 8.* **Privacy-accuracy tradeoff for DP-TTA methods with varying noise level.** We measure top-1 accuracy (%) (with standard deviation over 5 repeated runs) over 15 common corruption types under different privacy budgets ($\varepsilon, \delta = 1e^{-6}$), when adapting ImageNet-C at severity level 5 in the **continual** setting.

| Method | $\varepsilon$-DP | Noise | | | Blur | | | | Weather | | | | Digital | | | | Avg. |
|---|---|---|---|---|---|---|---|---|---|---|---|---|---|---|---|---|---|
| | | Gauss. | Shot | Impul. | Defoc. | Glass | Motion | Zoom | Snow | Frost | Fog | Brit. | Contr. | Elastic | Pixel | JPEG | |
| Source ($f_{\theta_0}$) | $\infty$ | 53.9 | 53.3 | 54.1 | 49.6 | 32.3 | 52.3 | 45.4 | 60.2 | 62.4 | 65.8 | 77.3 | 36.6 | 45.2 | 67.3 | 69.3 | 55.0 (0.02) |
| Tent | $\infty$ | 55.8 | 58.3 | 59.3 | 53.0 | 46.7 | 59.1 | 53.2 | 63.0 | 61.4 | 66.8 | 78.1 | 64.3 | 53.1 | 69.4 | 70.3 | 60.8 (0.01) |
| Tent + clip | $\infty$ | 60.9 | 63.4 | 63.4 | 56.7 | 56.9 | 62.8 | 59.8 | 66.2 | 66.1 | 71.1 | 78.1 | 62.9 | 62.8 | 71.7 | 70.8 | 64.9 (0.05) |
| DP-Tent | 20 | 58.9 | 62.1 | 62.3 | 54.3 | 54.2 | 60.9 | 57.4 | 64.0 | 64.0 | 68.4 | 78.0 | 59.8 | 58.6 | 70.6 | 70.1 | 62.9 (0.06) |
| | 15 | 57.9 | 61.2 | 61.7 | 54.0 | 52.7 | 60.7 | 56.6 | 64.2 | 62.9 | 67.9 | 78.1 | 61.1 | 59.1 | 70.4 | 70.5 | 62.6 (0.11) |
| | 10 | 57.8 | 61.0 | 61.6 | 53.7 | 52.5 | 60.4 | 56.1 | 63.7 | 62.4 | 67.1 | 77.9 | 59.1 | 58.1 | 70.0 | 70.1 | 62.1 (0.11) |
| | 5 | 56.1 | 58.3 | 59.5 | 52.7 | 46.2 | 58.7 | 52.9 | 63.4 | 62.2 | 66.7 | 78.1 | 61.8 | 55.2 | 69.2 | 70.6 | 60.8 (0.14) |
| | 1 | 55.1 | 57.2 | 58.1 | 51.5 | 42.8 | 57.2 | 50.3 | 61.7 | 61.3 | 63.6 | 77.4 | 52.4 | 52.6 | 67.0 | 69.3 | 58.5 (0.32) |
| EATA | $\infty$ | 59.3 | 62.9 | 63.3 | 58.6 | 57.9 | 64.0 | 62.6 | 67.7 | 67.2 | 72.0 | 79.3 | 62.0 | 66.5 | 72.6 | 72.6 | 65.9 (0.07) |
| EATA + clip | $\infty$ | 60.3 | 63.7 | 63.6 | 57.0 | 58.2 | 63.7 | 62.6 | 67.9 | 67.1 | 72.0 | 78.4 | 64.3 | 66.8 | 73.1 | 71.5 | 66.0 (0.04) |
| DP-EATA | 20 | 58.9 | 62.0 | 62.3 | 54.4 | 54.4 | 61.0 | 57.6 | 64.2 | 64.1 | 68.4 | 78.0 | 59.9 | 58.9 | 70.6 | 70.2 | 63.0 (0.06) |
| | 15 | 57.9 | 61.1 | 61.7 | 54.1 | 52.8 | 60.6 | 56.6 | 64.1 | 62.9 | 68.0 | 78.1 | 61.1 | 59.1 | 70.4 | 70.5 | 62.6 (0.09) |
| | 10 | 57.8 | 60.9 | 61.6 | 53.7 | 52.6 | 60.3 | 56.2 | 63.7 | 62.4 | 67.2 | 77.9 | 59.2 | 58.4 | 70.0 | 70.1 | 62.1 (0.08) |
| | 5 | 56.9 | 59.7 | 60.7 | 53.5 | 50.1 | 59.9 | 54.7 | 64.0 | 61.9 | 66.2 | 77.6 | 57.6 | 56.7 | 69.1 | 70.1 | 61.2 (0.26) |
| | 1 | 55.1 | 57.2 | 58.1 | 51.4 | 42.8 | 57.2 | 50.3 | 61.6 | 61.3 | 63.7 | 77.4 | 52.4 | 52.6 | 66.9 | 69.2 | 58.5 (0.31) |
| SAR | $\infty$ | 56.9 | 58.8 | 58.2 | 47.3 | 47.0 | 61.8 | 60.0 | 51.3 | 65.4 | 70.0 | 77.0 | 58.9 | 59.1 | 71.2 | 69.2 | 60.8 (0.14) |
| SAR + clip | $\infty$ | 57.5 | 59.3 | 59.7 | 53.7 | 52.4 | 57.9 | 55.3 | 59.9 | 60.8 | 64.1 | 75.9 | 58.2 | 56.0 | 67.7 | 68.4 | 60.5 (0.18) |
| DP-SAR | 20 | 57.0 | 58.6 | 58.7 | 52.1 | 48.7 | 56.6 | 52.6 | 61.4 | 60.9 | 62.9 | 76.3 | 56.5 | 56.9 | 66.6 | 68.2 | 59.6 (0.09) |
| | 15 | 56.9 | 58.5 | 58.6 | 51.9 | 48.5 | 56.4 | 52.4 | 61.2 | 60.8 | 62.3 | 76.2 | 55.5 | 56.6 | 66.6 | 68.1 | 59.4 (0.07) |
| | 10 | 55.6 | 57.7 | 58.3 | 51.7 | 45.2 | 57.6 | 51.0 | 61.8 | 60.8 | 64.6 | 77.6 | 59.0 | 53.3 | 67.7 | 68.9 | 59.4 (0.05) |
| | 5 | 55.5 | 57.6 | 58.2 | 51.8 | 44.7 | 57.3 | 50.5 | 61.5 | 60.5 | 64.0 | 77.3 | 56.9 | 53.2 | 67.3 | 68.7 | 59.0 (0.10) |
| | 1 | 54.8 | 56.6 | 57.3 | 51.0 | 41.4 | 56.2 | 48.7 | 60.3 | 59.6 | 62.4 | 77.0 | 50.5 | 51.0 | 66.1 | 68.1 | 57.4 (0.13) |
| DeYO | $\infty$ | 57.1 | 59.0 | 59.1 | 53.6 | 52.1 | 58.2 | 54.4 | 62.4 | 61.8 | 65.5 | 76.6 | 60.1 | 58.9 | 68.1 | 68.5 | 61.0 (0.12) |
| DeYO + clip | $\infty$ | 56.2 | 59.1 | 58.7 | 51.9 | 54.7 | 60.1 | 57.5 | 64.5 | 64.1 | 69.7 | 76.9 | 60.7 | 65.5 | 71.4 | 69.0 | 62.7 (0.36) |
| DP-DeYO | 20 | 55.1 | 57.0 | 57.6 | 50.8 | 42.9 | 56.7 | 49.7 | 61.0 | 60.9 | 64.5 | 77.4 | 57.6 | 52.4 | 66.4 | 69.2 | 58.6 (0.04) |
| | 15 | 55.8 | 57.2 | 57.6 | 50.7 | 44.3 | 56.3 | 49.6 | 60.2 | 60.8 | 63.1 | 76.4 | 57.6 | 53.2 | 63.9 | 67.8 | 58.3 (0.17) |
| | 10 | 55.2 | 57.0 | 57.6 | 50.8 | 42.9 | 56.7 | 49.6 | 60.9 | 60.8 | 64.4 | 77.3 | 57.2 | 52.3 | 66.1 | 68.9 | 58.5 (0.05) |
| | 5 | 55.7 | 57.1 | 57.6 | 50.9 | 43.9 | 56.3 | 49.8 | 60.1 | 60.4 | 62.6 | 76.3 | 56.9 | 53.0 | 64.1 | 67.7 | 58.2 (0.16) |
| | 1 | 55.2 | 56.9 | 57.3 | 50.6 | 41.3 | 55.7 | 49.3 | 59.2 | 59.7 | 61.7 | 76.8 | 51.4 | 50.5 | 65.6 | 68.0 | 57.3 (0.05) |
| DeYO-COME | $\infty$ | 61.7 | 63.7 | 63.7 | 58.7 | 58.2 | 63.9 | 60.1 | 68.3 | 67.2 | 72.6 | 79.0 | 66.3 | 67.7 | 73.4 | 71.7 | 66.4 (0.10) |
| DeYO-COME + clip | $\infty$ | 62.0 | 64.2 | 63.8 | 58.5 | 59.1 | 65.1 | 63.9 | 70.0 | 68.6 | 74.0 | 78.9 | 66.9 | 70.3 | 74.6 | 72.1 | 67.5 (0.08) |
| DP-DeYO-COME | 20 | 60.7 | 62.9 | 62.8 | 55.2 | 54.3 | 62.0 | 59.1 | 67.1 | 66.1 | 71.6 | 78.6 | 63.1 | 63.7 | 72.5 | 71.0 | 64.7 (0.12) |
| | 15 | 60.1 | 62.6 | 62.7 | 53.4 | 52.2 | 60.9 | 57.4 | 66.4 | 65.6 | 71.4 | 78.8 | 63.1 | 61.4 | 72.4 | 70.8 | 63.9 (0.08) |
| | 10 | 60.1 | 62.4 | 62.6 | 53.1 | 51.6 | 60.6 | 57.2 | 66.1 | 65.2 | 71.0 | 78.6 | 61.9 | 61.0 | 72.0 | 70.6 | 63.6 (0.13) |
| | 5 | 59.7 | 62.0 | 62.0 | 51.2 | 50.4 | 58.5 | 55.6 | 63.6 | 63.6 | 67.8 | 77.8 | 55.5 | 59.4 | 70.8 | 69.3 | 61.8 (0.25) |
| | 1 | 56.2 | 60.0 | 60.9 | 44.9 | 38.3 | 51.9 | 50.3 | 61.2 | 60.2 | 63.9 | 77.5 | 55.1 | 50.9 | 69.8 | 68.8 | 58.0 (0.16) |

levels. Notably, TTA+clip methods show better performance than the not-clipped baseline with $1.9 - 14\%$ improvements in accuracy. Table 13 and Table 14 show the accuracy of the baseline adaptation methods and their clipping and DP variants at different privacy levels under continual and episodic adaptation, respectively. We observe a similar pattern: DP-Tent outperforms the Tent baseline at several privacy levels. Moreover, per-sample gradient clipping consistently improves the accuracy of both Tent and EATA in both continual and episodic settings, with gains of $1\%$–$5\%$.

### B.4. Statistical Variance in Adaptation Performance

For the main experiments on ImageNet-C with a ViT source model, we repeat each experiment five times with different random seeds and report the standard deviation of the average accuracy across the 15 corruption types. Standard deviation values are shown in brackets in Table 8 and Table 9, and as shaded regions in Figure 1 and Figure 3. We also report standard deviations for the ablation experiments in Table 2 and Table 7. Overall, the TTA baselines, TTA+clip, and DP-TTA methods show relatively low variance across repeated runs, with most standard deviations across five runs remaining within $1\%$ accuracy.

We note an exceptional case under the episodic adaptation setting, e.g., Figure 3 and Table 9, where we observe larger standard deviations for DP-Tent and DP-DeYO at $\epsilon = 10$, DP-SAR at $\epsilon = 10, 15$, and the DeYO-COME baseline. This increased variance is primarily caused by the "fog" and "contrast" corruption classes, which have substantial performance fluctuations over random seeds (e.g. between $23.3\%$ to $86.5\%$ for DP-DeYO at $\epsilon = 10$). In such cases, lowering learning rate would often lead to decreased but less variable accuracy, e.g. comparing to the best tuned DP-DeYO at $\epsilon = 10$ with learning rate 0.001 has average accuracy of $61.4\%$ with standard deviation 2.18, decreasing learning rate to 0.0005 and 0.0001 lead to mean accuracy of $59.6\%, 56.1\%$ with standard deviation $1.73, 0.84$ respectively.

*Table 9.* **Privacy-accuracy tradeoff for DP-TTA methods with varying noise level.** We measure top-1 accuracy (%) (and standard deviation over 5 repeated runs) over 15 common corruption types under different privacy budgets ($\varepsilon, \delta = 1e^{-6}$), when adapting ImageNet-C at severity level 5 in the **episodic** setting.

| Method | $\varepsilon$-DP | Noise | | | Blur | | | | Weather | | | | Digital | | | | Avg. |
|---|---|---|---|---|---|---|---|---|---|---|---|---|---|---|---|---|---|
| | | Gauss. | Shot | Impul. | Defoc. | Glass | Motion | Zoom | Snow | Frost | Fog | Brit. | Contr. | Elastic | Pixel | JPEG | |
| Source ($f_{\theta_0}$) | $\infty$ | 53.9 | 53.3 | 54.1 | 49.6 | 32.3 | 52.3 | 45.4 | 60.2 | 62.4 | 65.8 | 77.3 | 36.6 | 45.2 | 67.3 | 69.3 | 55.0 (0.02) |
| Tent | $\infty$ | 58.7 | 59.7 | 60.1 | 58.8 | 53.8 | 62.6 | 58.7 | 54.6 | 59.7 | 70.0 | 78.6 | 66.1 | 60.3 | 72.2 | 71.2 | 63.0 (0.21) |
| Tent + clip | $\infty$ | 60.8 | 62.5 | 62.2 | 61.1 | 60.0 | 66.7 | 64.9 | 71.0 | 68.8 | 73.8 | 79.8 | 67.2 | 71.8 | 76.2 | 73.5 | 68.0 (1.72) |
| DP-Tent | 20 | 59.0 | 60.7 | 60.2 | 58.0 | 57.2 | 63.5 | 61.3 | 68.4 | 66.3 | 70.4 | 78.7 | 62.6 | 68.8 | 74.6 | 72.1 | 65.4 (0.08) |
| | 15 | 57.8 | 59.9 | 59.2 | 56.3 | 56.2 | 62.4 | 60.1 | 67.6 | 65.4 | 69.4 | 78.0 | 60.2 | 68.2 | 74.0 | 71.3 | 64.4 (0.06) |
| | 10 | 58.7 | 60.0 | 59.9 | 57.6 | 52.6 | 61.9 | 58.0 | 66.2 | 64.1 | 59.9 | 78.4 | 62.0 | 63.9 | 72.6 | 71.4 | 63.1 (1.65) |
| | 5 | 57.9 | 59.4 | 59.4 | 56.5 | 52.1 | 60.9 | 56.7 | 65.4 | 63.3 | 57.8 | 78.0 | 60.2 | 63.1 | 72.1 | 70.9 | 62.2 (0.25) |
| | 1 | 55.6 | 56.5 | 57.0 | 53.2 | 40.8 | 57.1 | 50.4 | 61.4 | 56.8 | 41.7 | 77.6 | 52.8 | 49.7 | 68.9 | 69.5 | 56.6 (0.86) |
| EATA | $\infty$ | 60.3 | 61.5 | 61.7 | 59.9 | 58.3 | 65.4 | 63.4 | 69.1 | 67.1 | 72.1 | 79.3 | 64.3 | 69.3 | 74.7 | 72.7 | 66.6 (0.16) |
| EATA + clip | $\infty$ | 61.0 | 62.5 | 62.4 | 61.1 | 60.1 | 66.6 | 65.4 | 71.2 | 69.0 | 73.5 | 79.8 | 66.1 | 72.1 | 76.2 | 73.7 | 68.0 (0.03) |
| DP-EATA | 20 | 58.9 | 60.8 | 60.2 | 58.1 | 57.1 | 64.3 | 61.4 | 68.6 | 66.5 | 70.7 | 78.7 | 62.8 | 68.8 | 74.6 | 72.1 | 65.5 (0.09) |
| | 15 | 57.8 | 59.8 | 59.1 | 56.4 | 56.4 | 62.4 | 60.3 | 67.5 | 65.2 | 69.2 | 77.8 | 60.4 | 68.2 | 73.7 | 71.4 | 64.4 (0.10) |
| | 10 | 54.8 | 57.2 | 56.3 | 52.3 | 53.6 | 59.4 | 57.0 | 64.8 | 62.5 | 65.9 | 76.3 | 53.9 | 66.1 | 72.1 | 69.8 | 61.5 (0.14) |
| | 5 | 57.8 | 59.2 | 59.2 | 56.4 | 52.0 | 60.8 | 56.6 | 65.0 | 63.1 | 59.1 | 77.8 | 60.2 | 63.0 | 71.9 | 70.7 | 62.2 (0.22) |
| | 1 | 55.6 | 56.5 | 57.0 | 53.2 | 40.8 | 57.1 | 50.4 | 61.4 | 56.8 | 41.6 | 77.5 | 52.8 | 49.7 | 68.8 | 69.5 | 56.6 (0.88) |
| SAR | $\infty$ | 57.5 | 59.4 | 59.5 | 57.6 | 56.5 | 63.2 | 61.5 | 57.0 | 64.3 | 67.0 | 77.6 | 63.6 | 67.6 | 73.5 | 71.5 | 63.8 (0.02) |
| SAR + clip | $\infty$ | 57.5 | 58.6 | 58.8 | 57.2 | 53.0 | 61.0 | 57.5 | 64.7 | 62.3 | 67.4 | 78.1 | 62.4 | 61.9 | 71.5 | 70.8 | 62.9 (0.02) |
| DP-SAR | 20 | 54.2 | 56.1 | 55.3 | 53.1 | 52.2 | 58.7 | 56.8 | 64.9 | 62.2 | 67.4 | 76.6 | 58.5 | 66.1 | 71.8 | 69.7 | 61.6 (0.08) |
| | 15 | 57.1 | 58.3 | 58.4 | 56.4 | 51.3 | 60.2 | 56.2 | 62.9 | 61.5 | 62.4 | 78.1 | 61.2 | 60.7 | 71.6 | 70.8 | 61.8 (2.18) |
| | 10 | 55.8 | 57.5 | 57.3 | 55.1 | 50.9 | 59.1 | 55.8 | 63.9 | 62.1 | 63.1 | 77.5 | 60.0 | 62.5 | 70.6 | 70.0 | 61.4 (1.78) |
| | 5 | 54.4 | 56.4 | 56.2 | 53.5 | 50.0 | 57.5 | 54.2 | 62.9 | 61.3 | 61.9 | 76.9 | 58.1 | 61.6 | 69.7 | 69.5 | 60.3 (0.10) |
| | 1 | 55.1 | 56.1 | 56.4 | 53.2 | 41.1 | 56.5 | 49.9 | 61.3 | 56.8 | 44.2 | 77.4 | 52.7 | 50.3 | 68.0 | 69.7 | 56.6 (1.11) |
| DeYO | $\infty$ | 56.8 | 58.8 | 58.8 | 57.1 | 56.4 | 63.7 | 61.2 | 68.0 | 63.8 | 68.3 | 78.3 | 63.1 | 68.6 | 73.5 | 71.4 | 64.5 (0.14) |
| DeYO + clip | $\infty$ | 56.7 | 58.8 | 57.9 | 56.1 | 55.4 | 62.8 | 60.8 | 67.9 | 65.2 | 70.9 | 77.7 | 62.5 | 69.5 | 73.6 | 70.9 | 64.5 (1.0) |
| DP-DeYO | 20 | 55.8 | 57.1 | 56.7 | 55.2 | 50.8 | 59.3 | 55.9 | 63.4 | 61.4 | 66.1 | 77.4 | 60.4 | 62.4 | 70.5 | 69.7 | 61.5 (0.10) |
| | 15 | 55.7 | 57.0 | 56.6 | 55.1 | 50.7 | 59.2 | 55.8 | 63.4 | 61.4 | 66.2 | 77.4 | 60.1 | 62.3 | 70.5 | 69.7 | 61.4 (0.11) |
| | 10 | 55.9 | 57.3 | 56.9 | 55.1 | 50.8 | 59.3 | 55.7 | 63.3 | 61.2 | 65.9 | 77.3 | 59.7 | 62.5 | 70.5 | 69.9 | 61.4 (2.18) |
| | 5 | 54.7 | 56.3 | 55.5 | 53.7 | 50.1 | 58.1 | 54.2 | 62.4 | 60.5 | 65.6 | 76.8 | 56.8 | 61.8 | 69.9 | 69.4 | 60.4 (0.08) |
| | 1 | 54.2 | 54.4 | 54.8 | 51.4 | 33.5 | 53.2 | 46.3 | 60.4 | 61.5 | 61.5 | 77.4 | 44.5 | 45.9 | 67.5 | 69.4 | 55.7 (0.04) |
| DeYO-COME | $\infty$ | 61.9 | 63.2 | 63.0 | 61.6 | 60.7 | 67.1 | 65.6 | 71.8 | 69.0 | 74.2 | 80.1 | 68.1 | 71.9 | 76.5 | 73.9 | 68.6 (2.36) |
| DeYO-COME + clip | $\infty$ | 62.0 | 63.6 | 63.2 | 62.4 | 61.5 | 67.9 | 66.8 | 72.5 | 70.1 | 75.0 | 80.0 | 68.4 | 73.2 | 76.5 | 73.8 | 69.1 (0.04) |
| DP-DeYO-COME | 20 | 60.1 | 61.7 | 61.5 | 59.7 | 58.6 | 65.2 | 63.7 | 70.6 | 67.5 | 73.6 | 79.5 | 66.2 | 70.8 | 75.4 | 73.0 | 67.1 (0.01) |
| | 15 | 59.2 | 60.9 | 60.7 | 58.6 | 57.8 | 64.3 | 62.9 | 70.0 | 66.9 | 72.9 | 79.1 | 64.7 | 70.5 | 75.0 | 72.6 | 66.4 (0.04) |
| | 10 | 60.4 | 61.7 | 61.9 | 59.0 | 54.9 | 63.4 | 60.0 | 69.1 | 65.6 | 71.5 | 79.8 | 66.1 | 66.2 | 74.6 | 72.6 | 65.8 (0.04) |
| | 5 | 59.6 | 61.2 | 61.1 | 57.8 | 54.1 | 62.6 | 58.9 | 68.5 | 64.9 | 70.6 | 79.6 | 64.5 | 65.7 | 74.2 | 72.4 | 65.1 (0.06) |
| | 1 | 57.5 | 58.1 | 58.2 | 52.3 | 40.8 | 57.2 | 51.6 | 63.0 | 58.9 | 58.0 | 78.1 | 56.7 | 52.7 | 70.1 | 70.0 | 58.9 (0.11) |

## B.5. Hyperparameter Sensitivity and Privacy Accounting for Hyperparameter Tuning

We evaluate hyperparameter sensitivity and show the results are robust to the tuning method. Table 15 shows the result when tuning per-sample clipping threshold ($C$) and learning rate ($lr$) over a grid for two DP-TTA settings, DP-Tent in continual adaptation setting at $\epsilon = 10$, and DP-EATA in episodic adaptation setting at $\epsilon = 1$, when adapting on ImageNet-C with ViT source model. We skip entries (marked by "/") because smaller $C$ usually pairs well with larger learning rates and vice versa. For DP-Tent-Continual at $\epsilon = 10$, we observe that accuracy remains stable (within 58%–62%) for many hyperparameter combinations, but fails for some combinations (with accuracy $< 10\%$). Similarly, for DP-EATA-Episodic at $\epsilon = 1$, most results are within the range of 52%–57%, with some clear failing hyperparameter choices with accuracy $< 10\%$. We observe similar patterns for other DP-TTA hyperparameter search results. Overall, our DP TTA works at a range of values.

As privacy accounting for tuning is often a practical concern, we provide an estimate on the additional privacy cost for tuning DP-TTA methods. Taking the DP-Tent-Continual setting in Table 15 as a demonstration, we can tune our $\epsilon = 1$ model over $k = 14$ tunings by releasing the DP accuracy of each with additive Gaussian noise. We compute accuracy from $n = 750,000$ samples, and the $L2$-sensitivity under our change-one neighboring definition is $\Delta = 1/n \approx 2 \times 10^{-6}$.

*Table 10.* **Clean accuracy on ImageNet-1k validation.** We evaluate the source model and each DP-TTA method under three privacy budgets ($\varepsilon$), using the best DP-tuned hyperparameters selected per $\varepsilon$. This sanity check verifies that the DP settings do not substantially degrade performance on data without shift.

| Method | Source ($f_{\theta_0}$) | DP-TENT | | | DP-EATA | | | DP-SAR | | | DP-DeYO | | | DP-DeYO-COME | | |
|---|---|---|---|---|---|---|---|---|---|---|---|---|---|---|---|---|
| Privacy ($\varepsilon$) | – | $\infty$ | 1 | 5 | $\infty$ | 1 | 5 | $\infty$ | 1 | 5 | $\infty$ | 1 | 5 | $\infty$ | 1 | 5 |
| Clean Acc. (%) | 85.1 | 84.7 | 83.0 | 81.8 | 84.3 | 83.9 | 84.1 | 84.5 | 84.7 | 83.8 | 85.1 | 83.5 | 83.8 | 85.1 | 84.4 | 84.3 |

*Table 11.* **Effect of batch size on adaptation performance.** The results are mean top-1 accuracy (%) over 15 corruption types when adapting on ImageNet-C at severity level 5 under a **episodic** setting.

| Method | Batch Size | | | | | |
|---|---|---|---|---|---|---|
| | 1 | 8 | 16 | 32 | 64 | 128 |
| Tent | 49.8 | 63.1 | 62.8 | 63.2 | 63.0 | 62.1 |
| Tent + clip | 67.7 | 67.6 | 68.4 | 67.6 | 68.0 | 67.7 |
| DP-Tent($\epsilon = 20$) | 59.6 | 60.7 | 61.4 | 61.8 | 62.9 | 62.1 |
| DP-Tent($\epsilon = 15$) | 58.9 | 60.5 | 61.2 | 61.8 | 62.6 | 62.5 |
| DP-Tent($\epsilon = 10$) | 56.9 | 60.5 | 60.8 | 61.6 | 62.1 | 62.3 |
| DP-Tent($\epsilon = 5$) | 56.8 | 57.1 | 53.5 | 60.6 | 60.8 | 61.7 |
| DP-Tent($\epsilon = 1$) | 55.7 | 55.8 | 56.8 | 57.4 | 58.5 | 59.4 |
| EATA | 65.2 | 66.1 | 66.5 | 67.1 | 66.6 | 65.3 |
| EATA + clip | 68.0 | 68.2 | 68.2 | 68.2 | 68.0 | 66.9 |
| DP-EATA($\epsilon = 20$) | 58.7 | 55.3 | 61.3 | 64.8 | 65.5 | 66.3 |
| DP-EATA($\epsilon = 15$) | 56.3 | 58.4 | 58.7 | 64.4 | 64.4 | 65.8 |
| DP-EATA($\epsilon = 10$) | 56.7 | 57.2 | 61.2 | 59.3 | 61.5 | 64.3 |
| DP-EATA($\epsilon = 5$) | 56.8 | 56.7 | 60.4 | 58.1 | 62.2 | 63.1 |
| DP-EATA($\epsilon = 1$) | 55.7 | 55.7 | 55.5 | 55.8 | 56.6 | 58.0 |
| SAR | 64.3 | 64.4 | 64.3 | 63.6 | 63.8 | 61.9 |
| SAR + clip | 63.6 | 64.7 | 64.2 | 63.1 | 62.9 | 60.3 |
| DP-SAR($\epsilon = 20$) | 59.1 | 60.5 | 61.2 | 61.4 | 61.6 | 62.4 |
| DP-SAR($\epsilon = 15$) | 58.5 | 56.5 | 60.2 | 61.6 | 61.8 | 61.8 |
| DP-SAR($\epsilon = 10$) | 56.6 | 59.3 | 59.0 | 60.4 | 61.4 | 61.8 |
| DP-SAR($\epsilon = 5$) | 55.8 | 55.7 | 58.8 | 59.5 | 60.3 | 60.6 |
| DP-SAR($\epsilon = 1$) | 55.5 | 55.7 | 56.2 | 55.6 | 56.6 | 57.0 |

*Table 12.* **Privacy-accuracy tradeoff for DP-TTA methods with varying noise level.** We measure top-1 accuracy (%) under different privacy budgets $(\varepsilon, \delta = 1e^{-6})$, when adapting **ViT** on ImageNet-R.

| Method | Privacy ($\varepsilon$) | | | | | | |
|---|---|---|---|---|---|---|---|
| | $\infty$ | $\infty$(clip) | 1 | 5 | 10 | 15 | 20 |
| Source ($f_{\theta_0}$) | 59.2 | - | - | - | - | - | - |
| Tent | 54.2 | 68.6 | 61.2 | 64.4 | 66.5 | 66.8 | 67.0 |
| EATA | 63.9 | 67.4 | 61.2 | 64.3 | 66.7 | 67.6 | 68.1 |
| SAR | 62.1 | 64.0 | 61.3 | 64.1 | 64.3 | 65.1 | 65.7 |
| DeYO | 63.7 | 67.6 | 61.4 | 64.8 | 65.2 | 65.4 | 66.0 |
| DeYO-COME | 63.6 | 67.3 | 59.2 | 60.7 | 61.4 | 61.7 | 61.8 |

Adding Gaussian noise with a scale of $\sigma = 0.01$ (the accuracy varies within $\pm 0.03$ with 99% probability) gives a privacy cost of $\epsilon = \Delta \times \sqrt{2 \ln(1.25/\delta)}/\sigma \approx 0.0011$ for $\delta = 1 \times 10^{-6}$. This is small comparing to the privacy cost of $\epsilon = 1$ for each run. The main privacy cost is fitting all $k$ models, which scales as $\sqrt{k}$ through DP composition. However, existing work (Liu & Talwar, 2018; Papernot & Steinke, 2022) provides a better analysis of this process, showing that the total privacy cost of fitting and selecting DP models can be as low as $2\times$-$3\times$ the cost of fitting and evaluating one model.

We further evaluate whether hyper-parameters tuned on a different distribution transfer to ImageNet-C. Since ImageNet-R contains far fewer samples than ImageNet-C, we simulate the continual adaptation setting by running adaptation over ImageNet-R for five consecutive passes when tuning hyper-parameters. We then apply the selected hyper-parameters to ImageNet-C. For Tent, EATA, and SAR, the tuned hyper-parameters largely transfer well: under small privacy budgets, the resulting ImageNet-C accuracy decreases only mildly compared with tuning directly on ImageNet-C. The main exception is at $\epsilon = 5$, where the gap is more noticeable. This suggests that DP-TTA performance is not highly sensitive to the exact tuning distribution, and that a small auxiliary dataset such as ImageNet-R can provide a practical alternative for hyperparameter selection.

*Table 13.* **Privacy-accuracy tradeoff for DP-TTA methods with varying noise level.** We measure top-1 accuracy (%) over 15 common corruption types under different privacy budgets $(\varepsilon, \delta = 1e^{-6})$, when adapting **ConvNeXt_Tiny** on ImageNet-C at severity level 5 in the **continual** setting.

| Method | $\varepsilon$-DP | Noise | | | Blur | | | | Weather | | | | Digital | | | | Avg. |
|---|---|---|---|---|---|---|---|---|---|---|---|---|---|---|---|---|---|
| | | Gauss. | Shot | Impul. | Defoc. | Glass | Motion | Zoom | Snow | Frost | Fog | Brit. | Contr. | Elastic | Pixel | JPEG | |
| Source ($f_{\theta_0}$) | $\infty$ | 41.6 | 41.1 | 41.5 | 35.4 | 15.9 | 37.1 | 37.2 | 43.4 | 53.2 | 45.0 | 73.3 | 41.9 | 25.2 | 56.3 | 60.6 | 43.2 |
| Tent | $\infty$ | 43.2 | 47.1 | 48.8 | 39.1 | 34.4 | 45.5 | 46.6 | 50.5 | 53.6 | 60.5 | 74.1 | 55.1 | 7.6 | 2.6 | 0.2 | 40.6 |
| Tent + clip | $\infty$ | 41.6 | 44.8 | 46.8 | 32.5 | 23.5 | 40.1 | 42.6 | 47.7 | 50.8 | 59.5 | 72.7 | 50.1 | 9.2 | 48.5 | 58.6 | 45.6 |
| DP-Tent | 20 | 44.7 | 48.8 | 50.5 | 37.3 | 37.1 | 46.3 | 49.2 | 52.4 | 56.5 | 59.0 | 73.8 | 54.4 | 41.8 | 59.4 | 63.4 | 51.6 |
| | 15 | 44.7 | 48.7 | 50.3 | 36.9 | 37.2 | 46.0 | 49.1 | 52.3 | 56.5 | 58.9 | 73.6 | 53.9 | 40.9 | 59.6 | 63.2 | 51.5 |
| | 10 | 44.5 | 48.5 | 50.1 | 35.5 | 34.9 | 45.4 | 48.8 | 51.9 | 56.2 | 58.4 | 73.3 | 52.9 | 40.5 | 59.2 | 63.1 | 50.9 |
| | 5 | 43.4 | 46.9 | 48.1 | 32.3 | 33.7 | 43.3 | 46.7 | 50.2 | 55.2 | 57.9 | 72.9 | 50.0 | 36.3 | 57.9 | 63.6 | 49.2 |
| | 1 | 41.6 | 41.2 | 41.6 | 36.1 | 16.1 | 37.2 | 37.8 | 43.8 | 52.5 | 45.4 | 73.2 | 42.3 | 25.6 | 56.7 | 60.9 | 43.5 |
| EATA | $\infty$ | 48.3 | 50.9 | 51.6 | 43.8 | 42.0 | 50.1 | 52.6 | 57.9 | 59.5 | 64.6 | 76.3 | 52.4 | 55.2 | 63.9 | 66.9 | 55.7 |
| EATA + clip | $\infty$ | 48.4 | 52.6 | 53.2 | 39.3 | 45.4 | 49.1 | 53.3 | 58.0 | 60.5 | 65.1 | 75.8 | 57.0 | 59.8 | 65.9 | 66.8 | 56.7 |
| DP-EATA | 20 | 44.7 | 48.8 | 50.3 | 37.2 | 37.4 | 46.3 | 49.1 | 52.2 | 56.4 | 58.7 | 73.2 | 53.9 | 41.9 | 59.0 | 63.0 | 51.5 |
| | 15 | 44.6 | 48.6 | 50.2 | 36.8 | 37.4 | 46.0 | 49.0 | 52.1 | 56.4 | 58.6 | 73.1 | 53.4 | 40.9 | 59.1 | 62.9 | 51.3 |
| | 10 | 44.5 | 48.4 | 50.0 | 35.4 | 35.2 | 45.3 | 48.7 | 51.7 | 56.0 | 58.1 | 72.8 | 52.4 | 40.6 | 58.7 | 62.8 | 50.7 |
| | 5 | 43.4 | 46.8 | 48.0 | 32.2 | 33.9 | 43.2 | 46.6 | 50.1 | 55.2 | 57.7 | 72.6 | 49.7 | 36.3 | 57.6 | 63.2 | 49.1 |
| | 1 | 41.6 | 41.2 | 41.6 | 36.1 | 16.1 | 37.2 | 37.8 | 43.8 | 52.5 | 45.4 | 73.2 | 42.4 | 25.6 | 56.7 | 60.9 | 43.5 |

*Table 14.* **Privacy-accuracy tradeoff for DP-TTA methods with varying noise level.** We measure top-1 accuracy (%) over 15 common corruption types under different privacy budgets $(\varepsilon, \delta = 1e^{-6})$, when adapting **ConvNeXt_Tiny** on ImageNet-C at severity level 5 in the **episodic** setting.

| Method | $\varepsilon$-DP | Noise | | | Blur | | | | Weather | | | | Digital | | | | Avg. |
|---|---|---|---|---|---|---|---|---|---|---|---|---|---|---|---|---|---|
| | | Gauss. | Shot | Impul. | Defoc. | Glass | Motion | Zoom | Snow | Frost | Fog | Brit. | Contr. | Elastic | Pixel | JPEG | |
| Source ($f_{\theta_0}$) | $\infty$ | 41.6 | 41.1 | 41.5 | 35.4 | 15.9 | 37.1 | 37.2 | 43.4 | 53.2 | 45.0 | 73.3 | 41.9 | 25.2 | 56.3 | 60.6 | 43.2 |
| Tent | $\infty$ | 46.3 | 47.5 | 48.1 | 45.1 | 5.4 | 48.5 | 36.9 | 53.9 | 36.1 | 61.5 | 76.3 | 58.0 | 1.8 | 63.6 | 65.8 | 46.3 |
| Tent + clip | $\infty$ | 46.6 | 48.1 | 48.4 | 43.7 | 7.1 | 47.8 | 43.8 | 54.8 | 53.9 | 60.6 | 76.6 | 55.9 | 1.6 | 64.6 | 67.3 | 48.1 |
| DP-Tent | 20 | 44.7 | 46.3 | 46.6 | 41.4 | 17.3 | 45.4 | 45.1 | 50.9 | 49.3 | 57.2 | 75.7 | 53.9 | 3.0 | 62.6 | 65.9 | 47.0 |
| | 15 | 44.7 | 46.3 | 46.5 | 41.4 | 17.6 | 45.4 | 45.0 | 50.9 | 49.2 | 57.2 | 75.6 | 53.8 | 3.0 | 62.6 | 65.9 | 47.0 |
| | 10 | 44.5 | 46.3 | 46.4 | 41.2 | 18.3 | 45.2 | 44.9 | 50.7 | 49.0 | 57.1 | 75.6 | 53.6 | 3.0 | 62.5 | 65.7 | 46.9 |
| | 5 | 43.9 | 45.8 | 45.9 | 40.1 | 20.6 | 44.2 | 44.2 | 50.1 | 46.0 | 56.6 | 75.3 | 52.4 | 3.1 | 62.0 | 65.4 | 46.4 |
| | 1 | 40.5 | 41.9 | 42.2 | 37.9 | 19.8 | 38.7 | 39.9 | 43.6 | 43.0 | 47.9 | 73.4 | 46.6 | 20.5 | 58.8 | 62.1 | 43.8 |
| EATA | $\infty$ | 48.3 | 49.7 | 50.2 | 44.4 | 41.8 | 49.9 | 51.1 | 56.8 | 57.3 | 63.6 | 76.8 | 39.9 | 52.3 | 64.1 | 66.3 | 54.2 |
| EATA + clip | $\infty$ | 48.4 | 49.5 | 50.3 | 42.5 | 41.0 | 49.9 | 51.8 | 57.9 | 57.8 | 62.2 | 76.2 | 53.3 | 58.7 | 66.2 | 67.4 | 55.6 |
| DP-EATA | 20 | 44.7 | 46.3 | 46.6 | 41.5 | 17.6 | 45.4 | 45.0 | 50.9 | 49.3 | 57.1 | 75.6 | 53.9 | 3.1 | 62.6 | 65.9 | 47.0 |
| | 15 | 44.6 | 46.3 | 46.5 | 41.4 | 17.9 | 45.4 | 45.0 | 50.9 | 49.3 | 57.1 | 75.5 | 53.8 | 3.1 | 62.6 | 65.8 | 47.0 |
| | 10 | 44.5 | 46.2 | 46.4 | 41.2 | 18.4 | 45.2 | 44.9 | 50.7 | 49.0 | 57.0 | 75.5 | 53.6 | 3.1 | 62.5 | 65.7 | 46.9 |
| | 5 | 43.9 | 45.7 | 45.9 | 40.2 | 20.8 | 44.2 | 44.1 | 50.0 | 46.2 | 56.5 | 75.3 | 52.5 | 3.1 | 62.0 | 65.4 | 46.4 |
| | 1 | 40.5 | 41.9 | 42.2 | 37.9 | 19.8 | 38.8 | 39.9 | 43.6 | 43.0 | 47.9 | 73.4 | 46.5 | 20.9 | 58.8 | 62.1 | 43.8 |

## B.6. Effect of Batch Size on TTA and DP-TTA Performances

We report the average accuracy over 15 corruption types in an episodic setting with ImageNet-C and ViT source model, when changing the batch size $B_t$ of the test data. Table 11 shows that the performance of DP-TTA methods are reasonably stable across batch sizes. Smaller batch sizes, e.g. $1, 8, 16, 32$, perform worse due to more noisy DP updates, though do not collapse, even at the extreme of batch size 1. Larger batch sizes mean less noisy updates, improving performance, though this effect competes with slower adaptation, which can decrease performance as shown in Tent, EATA, SAR and their +clip alternatives. We observe that there can be a crossover between adaptation benefit and noise accumulation when varying batch sizes. For example, as batch size decreases, SAR and SAR + clip suffer more from noise accumulation due to the larger number of adaptation steps, whereas larger batch sizes reduce this effect for DP-SAR. In the large-batch regime of 128, DP-SAR at $\epsilon = 20$ surpasses the non-private SAR baseline. We note that the best learning rate decreases with batch size, with for instance the best learning rate for DP-EATA($\epsilon = 1$) at batch size $128/64/32/16/8/1$ being $1 \times 10^{-3}/10^{-3}/10^{-4}/10^{-4}/10^{-4}/10^{-5}$, respectively, partially compensating for the increase in noise size compared to gradient size.

*Table 15.* **Hyperparameter sensitivity** for average accuracy over 15 corruption types, when tuning the per-sample clipping threshold ($C$) and learning rate ($lr$) over a grid, with skipped entries (marked by "/") because smaller $C$ usually pairs well with larger learning rates and vice versa. DP-TTA methods can work at a range of values.

| | C\lr | 0.0001 | 0.0005 | 0.001 | 0.005 | 0.01 | 0.05 | 0.1 | 0.5 |
|---|---|---|---|---|---|---|---|---|---|
| **DP-Tent-Continual** ($\epsilon = 10$) | 0.1 | / | / | / | 59.8 | 60.9 | 62.1 | 60.7 | 8.4 |
| | 1 | / | / | 60.9 | 61.9 | 60.0 | 8.3 | 2.0 | / |
| | 10 | 60.3 | 61.0 | 58.2 | 5.9 | / | / | / | / |
| **DP-EATA-Episodic** ($\epsilon = 1$) | C\lr | 0.0001 | 0.0005 | 0.001 | 0.005 | 0.01 | 0.05 | 0.1 | 0.5 |
| | 0.1 | / | / | / | 56.0 | 56.6 | 52.2 | 10.6 | 1.8 |
| | 1 | / | / | 56.5 | 53.1 | 34.7 | 1.8 | 0.9 | / |
| | 10 | 49.5 | 56.2 | 28.1 | 2.1 | / | / | / | / |

*Table 16.* **Hyperparameter sensitivity** for tuning the per-sample clipping threshold ($C$) and learning rate ($lr$) on ImageNet-R and testing on ImageNet-C (continual setting). This verifies that in most cases the hyperparameters generalize across shifts, so that it is possible to tune on separate data without any effect on the privacy of the test data.

| Method | Tune on IN-R | | | | | Tune on IN-C | | | | |
|---|---|---|---|---|---|---|---|---|---|---|
| | Privacy ($\varepsilon$) | | | | | Privacy ($\varepsilon$) | | | | |
| | 20 | 15 | 10 | 5 | 1 | 20 | 15 | 10 | 5 | 1 |
| Tent | 62.9 | 62.3 | 60.7 | 41.0 | 56.3 | 62.9 | 62.6 | 62.1 | 60.8 | 58.5 |
| EATA | 63.0 | 60.8 | 60.8 | 40.8 | 56.4 | 63.0 | 62.6 | 62.1 | 61.2 | 58.5 |
| SAR | 58.3 | 58.3 | 58.3 | 58.1 | 53.7 | 59.6 | 59.4 | 59.4 | 59.0 | 57.4 |

*Table 17.* **Hyperparameter sensitivity** for tuning per-sample clipping threshold ($C$) and learning rate ($lr$) on ImageNet-R and test on ImageNet-C (episodic setting).

| Method | Tune on IN-R | | | | | Tune on IN-C | | | | |
|---|---|---|---|---|---|---|---|---|---|---|
| | Privacy ($\varepsilon$) | | | | | Privacy ($\varepsilon$) | | | | |
| | 20 | 15 | 10 | 5 | 1 | 20 | 15 | 10 | 5 | 1 |
| Tent | 51.8 | 64.2 | 61.3 | 62.2 | 52.2 | 65.4 | 64.4 | 63.1 | 62.2 | 56.6 |
| EATA | 60.9 | 64.4 | 61.5 | 59.0 | 52.2 | 65.5 | 54.4 | 61.5 | 62.2 | 56.6 |
| SAR | 61.6 | 56.9 | 61.4 | 32.0 | 50.7 | 61.6 | 61.8 | 61.4 | 60.3 | 56.6 |

