# OpenReview forum: "Private and Stable Test-time Adaptation with Differential Privacy"
_ICML.cc/2026/Conference — ICML 2026 regular_

### Official Review · Reviewer_5mFP · 2026-02-20

**Soundness:** 2
**Presentation:** 3
**Significance:** 2
**Originality:** 3
**Overall Recommendation:** 5
**Confidence:** 4

**Summary:**

**Summary**

This paper addresses a critical vulnerability in Test-Time Adaptation (TTA): the privacy risk posed by dynamically updating model parameters on sensitive, unlabeled user data during inference. Because stateful TTA continuously modifies the model's weights based on incoming test streams, it inadvertently memorizes test data, making the system susceptible to privacy leaks (e.g., membership inference or data extraction attacks). To mitigate this, the authors introduce a framework for Differentially Private Test-Time Adaptation (DP-TTA), systematically converting several popular TTA algorithms (including Tent, EATA, SAR, DeYO, and COME) into mathematically private formulations.

The core methodology involves applying per-sample gradient clipping and injecting Gaussian noise into the gradient updates during the live inference stage. Theoretically, the authors provide a rigorous privacy analysis grounded in Gaussian Differential Privacy (GDP). They adopt a "change-one" neighboring definition suitable for streaming, sequential data (yielding a bounded l2-sensitivity of 2C ). They leverage the single-pass, online nature of TTA to show that the algorithm achieves strict privacy guarantees without suffering the compounding privacy budget decay typical of multi-epoch DP-SGD training.

The paper makes two primary contributions:

1. **Private and Effective Adaptation:** The authors empirically demonstrate on the ImageNet-C benchmark that DP-TTA algorithms successfully balance the privacy-utility tradeoff. Even under strict privacy budgets, DP-TTA successfully adapts to severe distribution shifts and significantly outperforms static, non-adapted source models.
2. **Algorithmic Stabilization via Clipping:** The paper uncovers a valuable empirical finding that per-sample gradient clipping, a mechanism strictly required for DP, independently acts as a powerful regularizer. By limiting the influence of individual, highly corrupted outlier images within a test batch, clipping alone improves the stability and overall accuracy of standard, non-private TTA methods.

**Compliance With Llm Reviewing Policy:**

Affirmed.

**Final Justification:**

I thank the authors for the thorough follow-up experiments addressing my remaining concerns. The cross-distribution tuning results (tuning on ImageNet-R, evaluating on ImageNet-C) demonstrate that the ~1% accuracy gap is robust and not an artifact of shared data structure. The comprehensive batch size sweep across DP-Tent and DP-EATA confirms that the framework degrades gracefully rather than collapsing at smaller batch sizes, and the SAR inversion analysis characterizes the noise accumulation tradeoff. Given these additional results and the overall strength of the contributions, I raise my score to 5.

**Key Questions For Authors:**

**1. Privacy Accounting for Hyperparameter Tuning**
You report tuning critical hyperparameters via a grid search directly on the test data. Under formal Differential Privacy guarantees, data-dependent hyperparameter selection constitutes a series of queries that require dividing the total privacy budget across the evaluation runs via composition theorems.

* **Question:** How do you justify this "oracle" parameter selection without mathematically accounting for the  budget spent during the search? Alternatively, can you provide results utilizing a formal private selection method (e.g., the Exponential Mechanism), or report the un-tuned performance across all values of  to demonstrate parameter stability? If you can mathematically justify the tuning process, provide a private selection mechanism, or demonstrate that the model remains robust across default values of C without relying on oracle selection, I will significantly raise my score.

**2. Statistical Variance and Standard Deviations**
The experimental results currently report only average accuracies. Both Test-Time Adaptation (prone to collapse from adverse batches) and Differential Privacy (which actively injects Gaussian noise) are inherently high-variance processes.

* **Question:** What is the standard deviation or confidence interval of the reported accuracies across multiple random seeds (particularly at strict privacy budgets)? Providing standard deviations will allow me to assess whether the performance improvements over the base model are statistically significant and reliable, or if they are heavily dependent on favorable noise draws. Including these metrics will resolve my primary concern on reproducibility.

**3. Robustness to Small Batch Sizes and Non-I.I.D. Streams**
The experimental design fixes the batch size to 64, mirroring idealized conditions. However, real-world TTA, especially in the private domains this paper targets, frequently operates on much smaller batch sizes.

* **Question:** How does the proposed DP-TTA framework perform when subjected to smaller batch sizes? Does the compounding variance of unstable batch statistics and DP noise injection cause model collapse? If you can provide empirical evidence (even on a subset of the ImageNet-C corruptions) demonstrating that the method does not catastrophically collapse under small batch sizes, it will improve my assessment of the paper's practical significance and real-world applicability.

**Limitations:**

yes

**Strengths And Weaknesses:**

### Strengths

**Originality:**
**Novel Problem Identification:** The paper introduces a highly original perspective by intersecting two previously disparate fields: Test-Time Adaptation (TTA) and Differential Privacy (DP). Recognizing that stateful, continuous test-time updates pose a significant privacy risek, is a novel and necessary insight for the field.
**Algorithmic Insights:** The discovery that per-sample gradient clipping acts as a powerful, independent regularizer that improves standard, non-private TTA performance is a creative and highly valuable empirical finding. It simplifies the algorithmic complexity of existing methods that previously relied on diverse loss-filtering techniques.

**Significance:**
**Practical Relevance:** Addressing the privacy vulnerabilities of TTA is a critical problem, particularly as ML models become more prevalent in private and sensitive domains like healthcare and edge computing.
**Foundation for Future Work:** The work provides a strong foundational framework (DP-TTA) that future researchers can build upon. The authors provide a framework for deploying adaptive models in environments bound by strict legal privacy constraints.

**Soundness:**
**Theoretical Adaptation:** The theoretical adaptation of DP-SGD to the streaming TTA setting is rigorous and mathematically sound. The authors correctly identify that the online, "one-and-done" nature of TTA eliminates the need for complex multi-epoch composition rules.
**Valid Mathematical Proofs:** The shift from a "leave-one-out" to a "change-one" neighboring definition for streaming data, and the resulting proofs bounding the sensitivity to , are technically correct and based on reasonable assumptions for online learning.

### Weaknesses
**Unaccounted Privacy Leakage via Hyperparameter Tuning:** The most critical flaw lies in the empirical evaluation of the privacy guarantees, specifically regarding data-dependent hyperparameter selection. The authors report tuning the l2 norm clipping threshold C and the learning rate via a grid search directly on the test data. Under formal DP, evaluating multiple hyperparameter configurations on private data requires dividing the total privacy budget across the evaluation runs via composition theorems. Understandably, the authors do not mathematically account for this budget expenditure, effectively reporting results as if the optimal parameters were known. This omission is severe because the algorithm's stability is mathematically highly sensitive to the choice of C . Because the authors' "change-one" neighboring definition yields an l2-sensitivity of 2C, the variance of the injected Gaussian noise scales directly with this bound. Consequently, assuming "oracle" access to the optimal overstates the model's empirical utility for a given epsilon. To resolve this without violating DP, the authors would need to either employ a private hyperparameter selection method (such as the Exponential Mechanism, which would consume a portion of the  budget) or transparently report the model's performance across all tested values of C to demonstrate parameter stability. By omitting both, safely deploying this method in practice without leaking privacy remains a major unsolved problem that the current empirical setup bypasses.

**Missing Variance Metrics:** Both TTA and DP are notoriously high-variance processes; standard TTA is prone to model collapse from poor batches, and DP actively corrupts gradients by injecting Gaussian noise. The authors only report average accuracies, entirely omitting standard deviations or error bars across multiple random seeds. This makes it impossible to mathematically assess the true stability of the DP-TTA updates or determine if the reported accuracies represent reliable expectations versus favorable noise draws.

**Batch Size Assumptions:** The experimental design strictly fixes the test data batch size to 64, mirroring the idealized laboratory conditions of the foundational TENT method. However, this assumption severely limits the practical applicability of the proposed method. In real-world deployment scenarios, TTA must frequently operate on continuous, heterogeneous data streams with smaller batch sizes (often N=1).  It is a well-documented failure mode in TTA literature that Batch Normalization-based adaptation collapses under small batch regimes because the empirical estimation of the batch mean and variance becomes statistically unreliable. By exclusively evaluating at a batch size of 64 the authors bypass this problem (the interaction between the inherently unstable batch statistics of small sample sizes and the degraded signal-to-noise ratio caused by injecting Differential Privacy Gaussian noise).

---

> ### Author Rebuttal · Authors · 2026-03-31
>
> We thank the reviewer for the constructive comments and address the key questions below. Due to the response time limit, our new experiments are on a subset of methods, corruptions, and privacy levels. We will include comprehensive experiments in the revision.
>
> > 1. Privacy Accounting for Hyperparameter Tuning
>
> We thank the reviewer for raising this practical concern. Privacy accounting for tuning is often omitted for DP training [[1](https://arxiv.org/abs/2204.13650), [2](https://arxiv.org/abs/1607.00133)], but we acknowledge its importance, and will add it to our discussion and impact statement. We now evaluate hyperparameter sensitivity and alternative tunings for privacy as requested and the results are robust to the tuning method.
>
> **Sensitivity** For DP-Tent in a continual setting at $\epsilon=1$, we show the accuracy for tuning the clipping $C$ and the learning rate $LR$ over a grid. We skip entries (marked by "/") because smaller $C$ usually pairs well with larger learning rates and vice versa.
>
> | C\LR | 0.00005 | 0.0001 | 0.0005 | 0.001 | 0.005 | 0.01 | 0.05 |
> |------|---------|--------|--------|-------|-------|------|------|
> | 0.1  | /       | /      | 50.6  |  51.5  |  52.6 | 53.5 |  7.9 |
> | 1    | /       |  48.3  | 49.1  | 50.1  |  7.6 | 1.8 | /    |
> | 10   | 47.4  |   50.6  |  4.7 |    1.4 | /     | /    | /    |
>
> The accuracy remains stable (within 47%–54%) for many settings, but fails for some combinations (with accuracy $<10\%$). Our DP TTA works at a range of values.
>
>
> **Private Tuning** We can tune our $\epsilon=1$ model over $k=14$ tunings by releasing the DP accuracy of each with additive Gaussian noise. We compute accuracy from $n=750,000$ samples, and the $L2$-sensitivity under our change-one neighbouring definition is $\Delta=1/n \approx 2\times10^{-6}$. Adding Gaussian noise with a scale of $\sigma=0.01$ (the accuracy varies within $\pm0.03$ with 99% probability) gives a privacy cost of $\epsilon=\Delta \times \sqrt{2\ln(1.25/\delta)} / \sigma \approx 0.0011$ for $\delta=1\times10^{-6}$. This is small vs. the privacy cost of $\epsilon=1$ for each run. The main privacy cost is fitting all $k$ models, which scales as $\sqrt{k}$ through DP composition. However, existing work [[3](https://arxiv.org/abs/1811.07971), [4](https://arxiv.org/abs/2110.03620)] provides a better analysis of this process, showing that the total privacy cost of fitting and selecting DP models can be as low as $2\times$-$3\times$ the cost of fitting and evaluating one model.
>
> **Different Tuning** To preserve privacy we tune to different data than the test data. The effect of tuning on held-out shifts vs. the test shifts is minimal and adaptation helps with either tuning.
>
> | epsilon | Held-out | Test |
> |---------|---------|-------|
> |      20 |    60.7 |  61.8 |
> |      15 |    60.4 |  61.1 |
> |      10 |    58.9 |  60.2 |
> |       5 |    57.5 |  58.4 |
> |       1 |    52.8 |  53.5 |
>
> We experiment on DP-Tent and select the best hyperparameters on the held-out shifts of IN-C, as recommended by [[5](https://arxiv.org/abs/2104.12928)], and then evaluate on the test data. The gap between "oracle" tuning on test vs. tuning on held-out shifts is negligible.
>
> > 2. Statistical Variance
>
> We report the mean accuracy and std. dev. of 3 seeds for Tent and Deyo-Come in baseline/clip/DP versions at $\epsilon=1,10,20$ in the continual setting (Tab. 8). Results are stable: the std. dev. is small vs. the mean. DP-TTA methods have higher variance at smaller $\epsilon$s (= more private) due to the added DP noise.
>
> | Method | Baseline | Clip | DP ($\epsilon=20$) | DP ($\epsilon=10$) | DP ($\epsilon=1$) |
> |---|---:|---:|---:|---:|---:|
> | Tent | 52.7$\pm$6.15 | 64.8$\pm$0.18 | 61.8$\pm$0.02 | 60.3$\pm$0.08 | 53.1$\pm$0.43 |
> | DeYo | 63.1$\pm$0.77 | 67.3$\pm$0.05 | 64.6$\pm$0.38 | 62.3$\pm$0.49 | 55.8$\pm$1.52 |
>
>
> > 3. Robustness to Small Batch Sizes
>
> We report the effect of smaller batch sizes. While accuracy varies, the change is not severe, and adaptation does not collapse even at the extreme of batch size = 1. We report the mean accuracy over 4 corruptions (impulse noise, zoom blur, brightness, jpeg compression = one from each family) for DP-Tent with batch sizes B=32 and 64 at $\epsilon=1,10,20$.
>
> |                        | B=32 | B=64 |
> |------------------------|------|------|
> | Tent                   | 57.4 | 60.8 |
> | Clip                   | 68.7 | 67.0 |
> | DP-Tent($\epsilon=20$) | 64.4 | 65.9 |
> | DP-Tent($\epsilon=10$) | 63.1 | 62.0 |
> | DP-Tent($\epsilon=1$)  | 54.4 | 54.9 |
>
> We evaluate B=1 with SAR/DP-SAR, because SAR evaluates B=1, and DP-SAR does not collapse.
>
> |                       | B=1  | B=64 |
> |-----------------------|------|------|
> | SAR                   | 63.7 | 59.6 |
> | SAR+Clip              | 64.9 | 62.9 |
> | DP-SAR($\epsilon=20$) | 57.3 | 62.9 |
> | DP-SAR($\epsilon=10$) | 56.0 | 59.4 |
> | DP-SAR($\epsilon=1$)  | 49.6 | 54.1 |

---

> > ### Author Rebuttal · Reviewer_5mFP · 2026-04-01
> >
> > I thank the authors for the additional experiments. The responses are a positive step, but I have the following follow-up questions.
> >
> > ### 1. Privacy Accounting
> >
> > - The held-out tuning uses ImageNet-C corruptions that share the same clean images and severity structure as the test corruptions. Does the reported 0.7--1.3% gap hold when tuning on a genuinely different distribution (e.g., ImageNet-R)?
> >
> > ### 2. Small Batch Sizes
> >
> > - Smaller batch sizes increase both the number of update steps and the per-step noise magnitude (since the fixed Gaussian noise is divided by |B_t| in Eq. 2). The B=1 experiment only uses SAR, which was designed for that regime. I understand that noise scaling with 1/B is inherent to any DP-SGD framework, so B=1 may be unrealistic for most DP-TTA methods. However, could you provide results at intermediate batch sizes (e.g., B=8 or B=16) for DP-Tent or DP-EATA to characterize the practical minimum batch size at which the framework provides meaningful adaptation?
> >
> > - Your own results reveal an inversion: SAR without DP gains +4.1% at B=1 vs B=64, but DP-SAR at epsilon=1 loses -4.5%. This is consistent with noise accumulating over 64x more update steps at B=1. Does this inversion scale predictably with the number of steps, and at what batch size does the crossover between adaptation benefit and noise accumulation occur?

---

> > > ### Author Response · Authors · 2026-04-07
> > >
> > > We thank the reviewer for the further detailed suggestions and report more results for tuning to different data and across more batch sizes.
> > >
> > > >Does the reported 0.7--1.3% gap hold when tuning on a genuinely different distribution (e.g., ImageNet-R)?
> > >
> > > ImageNet-C, continual, DP-Tent
> > > | epsilon | tuned on IN-R | tuned on held-out set | tuned on IN-C |
> > > |---|---|---|---|
> > > | 20 | 61.8 | 61.8 | 61.8 |
> > > | 15 | 61.1 | 61.1 | 61.1 |
> > > | 10 | 58.9 | 58.9 | 60.2 |
> > > | 5 | 58.5 | 58.5 | 58.5 |
> > > | 1 | 53.6 | 53.6 | 53.6 |
> > >
> > > ImageNet-C, episodic, DP-Tent
> > > | epsilon | tuned on IN-R | tuned on IN-C |
> > > |---|---|---|
> > > | 20 | 57.7 | 57.7 |
> > > | 15 | 57.3 | 58.3 |
> > > | 10 | 56.4 | 56.4 |
> > > | 5 | 55.8 | 55.8 |
> > > | 1 | 48.3 | 48.3 |
> > >
> > > As suggested, we tuned hyperparameters on ImageNet-R, and evaluated adaptation on ImageNet-C. ImageNet-R contains 30k images, which is much closer to the episodic length of ImageNet-C (50k images per corruption). Using the best ImageNet-R hyperparameters and testing on ImageNet-C leads to a ~1% difference relative to tuning directly on ImageNet-C.
> > >
> > > To better match the continual setting, we tuned adaptation on ImageNet-R for 5 consecutive passes without resetting to select the best values of C and learning rate. We then evaluated those hyperparameters in the continual setting on ImageNet-C. This produced almost the same performance as tuning directly on ImageNet-C. We did the same hyperparameter sweep and selection protocol on both the ImageNet-C hold-out set and ImageNet-R as in the main experiments. Overall, these results suggest that tuning on a different dataset can yield comparably strong performance, but of course this remains dependent on the dataset and shifts.
> > >
> > > >However, could you provide results at intermediate batch sizes (e.g., B=8 or B=16) for DP-Tent or DP-EATA to characterize the practical minimum batch size at which the framework provides meaningful adaptation?
> > >
> > > The following tables show results for DP-Tent and DP-EATA across batch size.
> > > | Tent | bs=1 | bs=8 | bs=16 | bs=32 | bs=64 |
> > > |---|---:|---:|---:|---:|---:|
> > > | baseline | 45.69 | 58.01 | 59.27 | 57.36 | 60.81 |
> > > | Tent+clip | 62.92 | 69.61 | 69.28 | 68.73 | 67.02 |
> > > | DP-Tent($\epsilon=20$) | 58.73 | 61.43 | 63.56 | 64.41 | 65.93 |
> > > | DP-Tent($\epsilon=15$) | 57.37 | 61.22 | 62.71 | 64.07 | 64.75 |
> > > | DP-Tent($\epsilon=10$) | 53.47 | 60.68 | 60.81 | 63.09 | 62.03 |
> > > | DP-Tent($\epsilon=5$) | 52.21 | 58.12 | 57.93 | 60.71 | 61.35 |
> > > | DP-Tent($\epsilon=1$) | 48.86 | 52.77 | 52.23 | 54.37 | 54.90 |
> > >
> > > | EATA | bs=1 | bs=8 | bs=16 | bs=32 | bs=64 |
> > > |---|---:|---:|---:|---:|---:|
> > > | baseline | 56.60 | 63.82 | 66.02 | 66.09 | 67.45 |
> > > | EATA+clip | 65.13 | 68.89 | 68.42 | 69.11 | 68.82 |
> > > | DP-EATA($\epsilon=20$) | 58.69 | 61.40 | 63.74 | 64.48 | 65.65 |
> > > | DP-EATA($\epsilon=15$) | 57.35 | 61.16 | 62.96 | 64.14 | 62.25 |
> > > | DP-EATA($\epsilon=10$) | 53.48 | 60.66 | 60.57 | 63.25 | 62.05 |
> > > | DP-EATA($\epsilon=5$) | 52.23 | 58.13 | 57.91 | 60.66 | 61.75 |
> > > | DP-EATA($\epsilon=1$) | 48.67 | 52.78 | 52.24 | 54.38 | 54.93 |
> > >
> > > The performance of DP models is reasonably stable across batch sizes. Smaller batch sizes perform worse due to more noisy DP updates, though do not collapse. Larger batch sizes mean less noisy updates, improving performance, though this effect competes with slower adaptation, which can decrease performance. Note that the best learning rate decreases with batch size, with for instance the best learning rate for DP-Tent($\epsilon=1$) at bs=64/32/16/8/1 being 0.01/0.01/0.005/0.001/0.0005, respectively, partially compensating for the increase in noise size compared to gradient size.
> > >
> > > >Does this inversion scale predictably with the number of steps, and at what batch size does the crossover between adaptation benefit and noise accumulation occur?
> > >
> > > | batch size | bs=1 | bs=8 | bs=64 | bs=128 | bs=256 | bs=512 |
> > > |---|---:|---:|---:|---:|---:|---:|
> > > | SAR | 63.7 | 64.7 | 59.6 | 60.5 | 55.8 | 48.6 |
> > > | DP-SAR($\epsilon=1$) | 49.6 | 52.2 | 54.1 | 55.8 | 55.8 | 57.5 |
> > >
> > > We further evaluated more batch sizes(B=1,8,64,128,256,512) and found that the trend is broadly consistent with the reviewer’s hypothesis. As batch size decreases, DP-SAR at ϵ=1 suffers more from noise accumulation due to the larger number of adaptation steps, whereas larger batch sizes reduce this effect. In the large-batch regime, DP-SAR becomes comparable to and then surpasses the non-private SAR baseline. This suggests that the inversion evolves in a consistent direction with batch sizes.

---

### Official Review · Reviewer_apFJ · 2026-02-25

**Soundness:** 3
**Presentation:** 2
**Significance:** 3
**Originality:** 3
**Overall Recommendation:** 5
**Confidence:** 4

**Summary:**

**Summary:**  This paper provides a general framework for differentially private (DP) test-time adaptation (TTA). During inference time, the data are assumed to be coming in batches, and a gradient descent step is taken using per-gradient clipping for each of the batches. Simulation studies show that the proposed algorithm enjoys a healthy privacy-accuracy trade-off.

**Compliance With Llm Reviewing Policy:**

Affirmed.

**Final Justification:**

My concerns have been adequately addressed. I have raised my score.

**Key Questions For Authors:**

- In Figure 1, the performance of DP-TENT and DP-DeYO-COME is better than the non-private baseline under a low-privacy regime. How is this possible?
 - Line 311, 1st column: The authors discuss the importance of per-gradient clipping in improving the performance of the proposed algorithm. However, as far as I understand, per-sample-gradient clipping may lead to more information loss than batch gradient clipping. Therefore, sample-level clipping could deliver inferior performance. What is the intuition behind its success in the experimental setups of the paper?

**Limitations:**

Yes

**Strengths And Weaknesses:**

**Strength:**
 - The algorithm is surprisingly simple and general. It can be adapted to many other TTA frameworks.
 - Algorithm 1 uses the batches only once during TTA. This helps to provide atighter bound on the privacy accounting.


**Weakness:**
 - There is no theoretical guarantee of the algorithm's utility.
 - Algorithm 1 essentially performs a single-epoch batch gradient descent. Therefore, it risks learning the parameter of interest incorrectly. Ideally, one should run multiple epochs, which may aggravate the privacy cost.

---

> ### Author Rebuttal · Authors · 2026-03-31
>
> We thank the reviewer for the constructive comments and address the weaknesses and key questions below.
>
> > There is no theoretical guarantee of the algorithm's utility
>
> True. At present differential Privacy (DP) has developed more theory than test-time adaptation (TTA). Neither the existing baseline nor our clipped or DP editions of the TTA methods studied in this work have theoretical guarantees for the accuracy of adaptation so we have not altered their current status in this regard. However, by incorporating DP into adaptation we have provided the first theoretical guarantees on the _privacy_ of the updates even if more theory on the _accuracy_ of the updates is still needed. Extending the theoretical guarantees of DP-TTA methods is an important direction of future work, and DP theory may further inform optimization for adaptation.
>
> > Ideally, one should run multiple epochs, which may aggravate the privacy cost
>
> Standard TTA focuses on streaming deployments in which only single epoch optimization is possible, e.g., Tent and EATA. That is, every data point is seen only once, and multiple epoch optimization is not possible. Running TTA for multiple epochs is considered its own special setting of "multiple visit" or "recurring" TTA [1, 2, 3]. If we were to extend to this newer multi-epochs setting, we would adapt privacy accounting through composition, and potentially with subsampling over the whole dataset to get better privacy/utility tradeoff with amplification results [4,5]. However, we leave this as future work, since we focus on first establishing differential privacy in the most common and established TTA setting of online and single epoch optimization.
>
> [1] Persistent Test-time Adaptation in Recurring Testing Scenarios. Hoang et al. NeurIPS'24.
>
> [2] ReservoirTTA. Vray* & Tomar* et al. NeurIPS'25.
>
> [3] DPCore. Zhang et al. ICML'25.
>
> [4] Abadi, Martin, et al. "Deep learning with differential privacy" (2016).
>
> [5] Balle, Borja, Gilles Barthe, and Marco Gaboardi. "Privacy amplification by subsampling: Tight analyses via couplings and divergences." (2018).
>
> > performance of DP-TENT and DP-DeYO-COME is better than the non-private baseline under a low-privacy regime
>
> This performance combines two effects: (1) an accuracy improvement due to per-sample gradient clipping (as shown in Section 4.2), and (2) an accuracy decrease due to adding DP noise. For methods such as DP-Tent and DP-DeYO-COME, it turns out that when the noise is small (i.e. in low privacy regime), the benefit from per-sample gradient clipping outweighs the negative effect of adding DP noise, therefore the overall performance improves and surpasses the non-private baseline.
>
> > per-sample-gradient clipping may lead to more information loss than batch gradient clipping
>
> Averaging per-sample clipped gradients can indeed introduce information loss (and bias in the aggregated gradient). However, our experiments with filtering (see Table 2), and the intuition for introducing filters in TTA in the first place [6 (the EATA baseline)], suggest that some examples can adversarially affect model performance. In such cases, per-batch clipping does not diminish the relative effect of those bad samples, while per-sample gradient clipping explicitly bounds those sample’s contribution to the model update, therefore reducing the sensitivity to such harmful samples (this is clipping’s role in ensuring DP too). Our best guess from current evidence is that this effect drives the performance improvements, though further isolating the effect remains an interesting question for future work!
>
> [6] Niu, Shuaicheng, et al. "Efficient test-time model adaptation without forgetting." ICML'24.

---

> > ### Author Rebuttal · Reviewer_apFJ · 2026-04-01
> >
> > My concerns have been adequately addressed.

---

> > > ### Author Response · Authors · 2026-04-07
> > >
> > > We thank the reviewer for their time and attention in reviewing this work and considering the rebuttal.

---

### Official Review · Reviewer_EdNe · 2026-03-09

**Soundness:** 3
**Presentation:** 3
**Significance:** 2
**Originality:** 2
**Overall Recommendation:** 5
**Confidence:** 4

**Summary:**

This paper studies a variety of test-time adaptation (TTA) techniques and shows how to adapt them to provide differential privacy to test samples by using gradient clipping and Gaussian noise.

**Compliance With Llm Reviewing Policy:**

Affirmed.

**Final Justification:**

I still think this paper passes threshold and will be of interest to a reasonable subset of ICML attendees.

**Key Questions For Authors:**

This is a nice little project that takes a comprehensive look at TTA methods and then applies the standard recipe of gradient clipping and Gaussian noise to guarantee differential privacy. Since I am not an expert on TTA I can really only provide some evaluation of the privacy component.

* From a privacy perspective, for TTA the assumption has to be that test-time instances come from individuals who do not appear in the training data for the original model. Is this a reasonable assumption in practice?
* Similarly, there are applications in which the test instances may be coming from the same individual. Consider a scenario where the original model is deployed within an organization (e.g. a school) or in a locality (e.g. a city) and the test instances are generated by students/residents. It's highly likely that a single individual will contribute several test instances. How can this framework be modified to account for this setting? Would a more sophisticated accounting method be needed?
* On the topic of accounting, the paper claims to "introduce adjustments to privacy accounting" but I do not see where this appears in the manuscript.
* Of course methods have to be evaluated on benchmark datasets with widely used models but is ViT + ImageNet the right choice for demonstrating the usefulness of these methods?

**Limitations:**

Yes.

**Strengths And Weaknesses:**

**Strengths:**

* This does appear to be the first paper on TTA under DP.
* Experimental results extensive and quite promising.

**Weaknesses:**

* From a DP methods perspective, there are not really new ideas here.

Overall, this paper is more or less correct: you can use the clipping + noise trick to do TTA too. Not all papers have to be surprising. The most interesting component is the fact that clipping helped even in the no-privacy regime, suggesting that existing TTA methods still have room for improvement.

---

> ### Author Rebuttal · Authors · 2026-03-31
>
> We thank the reviewer for their expertise on privacy and the constructive comments. We address the key questions:
>
> > [Privacy Assumption for Train vs. Test] From a privacy perspective, for TTA the assumption has to be that test-time instances come from individuals who do not appear in the training data
>
> We aim to protect the privacy of the test data, and not the training data, which we will reinforce beyond our statement on l. 74-75 by editing the abstract and introduction. The assumption that test instances come from individuals outside the original training set may not always hold, but it is still reasonable in many deployment scenarios where a pretrained model is adapted at test-time, on data from new users, clients, or environments. Even if there is some overlap, the DP guarantee still applies to the test samples used during adaptation.
>
> We note that we do not assume a private trained model, and our DP-TTA methods are compatible with either private or non-private training (Section 3.1, L158-159) within our scope to protect privacy with respect to test data.
>
> > applications in which the test instances may be coming from the same individual [...] a single individual will contribute several test instances
> > Would a more sophisticated accounting method be needed?
>
> To clarify: we consider an event-level DP setting, where privacy guarantees apply at the level of each data point, as is common for DP training [1]. Guarantees apply (at the record level) regardless of the number of contributions. This is the typical privacy setting, considered in most DP-SGD work (though not all). However, the concern raised in the review is valid: a user contributing many observations would incur a larger privacy loss at the user level. Addressing this issue would require enforcing the stronger user-level DP model. This would decrease utility, and be particularly challenging in a deployment setting of TTA where users are typically unknown and data points are visited only once, though it is an interesting avenue for future work!
>
> With the current threat model (event-level DP), the user-level privacy of a user contributing multiple observations over time degrades with group composition (graceful degradation, but fairly fast).
> It is very possible that some amplification by iteration effects [2] would make the real privacy much stronger. An analysis of such an effect in the TTA setting would be interesting future work, but is an open question beyond the scope of this paper.
>
> > Introduce Adjustments to Privacy Accounting
>
> Thank you for the close examination of privacy accounting in this work. To clarify, our adjustment _for adaptation_ is to change the neighbouring definition to be `change-one`, which is stronger than the `add/remove-one` definition that is typically used in DP-SGD work _for training_. We also apply accounting without amplification by subsampling as the model observes each test data once in the TTA setting we focus on in the paper. Finally, we adjust TTA algorithms when needed for DP. This helpful comment indicates that we can better explain this contribution as "identifying" and "customizing" the privacy accounting for the adaptation setting, rather than adjusting privacy accounting itself. We will edit L59-60 to "We identify and apply privacy accounting methods compatible with the streaming setting of TTA, and introduce adjustments to TTA algorithms to make common elements of TTA respect DP".
>
> > is ViT + ImageNet the right choice for demonstrating the usefulness
>
>
> Thank you for this question. Our main experiments focus on ViT-B/16 and ImageNet-C, as ViT is a widely used backbone in recent TTA studies and ImageNet-C is a commonly used benchmark in this literature. We agree that evaluating these methods on a broader range of models and datasets is important for demonstrating their usefulness more fully. To provide a more comprehensive evaluation, we additionally include rebuttal experiments on a new dataset, ImageNet-R, and on a model from a different family, ConvNeXt.
>
> Please see our response to pR5g under "[more datasets and models](https://openreview.net/forum?id=Ct0HIcLIMX&noteId=9987ty8cSq)" for rebuttal experiments with another dataset (IN-R) and another model (ConvNeXt) which show the same type of results as we report with IN-C and ViT-B.
>
>
> [1] Abadi, Martin, et al. "Deep learning with differential privacy." (2016).
>
> [2] Feldman, Vitaly, et al. "Privacy amplification by iteration." (2018).

---

> > ### Author Rebuttal · Reviewer_EdNe · 2026-03-31
> >
> > Thank you for the clarifications. I think that my questions have been answered and think the paper should be accepted but even with additional experiments the connection between the privacy applications and the specific experiments is less than clear. This is not as bad as, e.g. writing a paper about fairness and evaluating it on MNIST but it would be better to have a more compelling instance for privacy.

---

> > > ### Author Response · Authors · 2026-04-07
> > >
> > > Thank you for understanding the need to evaluate on common benchmarks for TTA for comparability and for informing that community. We agree that more private data, such as faces or medical imagery, would enable privacy-relevant experiments. Making such a private adaptation benchmark is an important direction for future work.

---

### Official Review · Reviewer_pR5g · 2026-03-11

**Soundness:** 3
**Presentation:** 4
**Significance:** 4
**Originality:** 3
**Overall Recommendation:** 5
**Confidence:** 4

**Summary:**

This paper analyzes how to adapt existing test-time adaptations (TTAs) into differential privacy (DP). The high-level idea of the approach is similar to DP-SGD; therefore, it first clips the per-sample gradients and then adds Gaussian noise for all the updates. Moreover, it further exploits the preprocessing property of DP to privatize some of the methods,
For each of the analysis types of TTA, a detailed description is given on how to practically adapt and privatize these methods by modifying the training procedure.

The work further claims that naively adapting TTA methods with DP makes them unstable and less effective, due to the small batch sizes and the data-dependent filtering and dynamic reweighting.

The paper focuses on five types of TTAs (Tent, EATA, SAR, DeYO, and COME) and evaluates their methods on imagenet-C on ViT-Base/16.

**Compliance With Llm Reviewing Policy:**

Affirmed.

**Final Justification:**

The rebuttal adequately addressed my concerns through additional experiments and clarifications, so I maintain my accept recommendation.

**Key Questions For Authors:**

See above.

**Limitations:**

Yes

**Strengths And Weaknesses:**

## Strengths
1. The paper is well written, easy to follow, and well motivated.
2. It shows that per-sample gradient clipping improves the utility of the adaptations.
3. It provides an interesting comparison between different alternatives to per-sample gradient clipping to limit the influence, such as batch-level clipping, filtering by the gradient norm, and filtering based on the loss, and shows that per-sample clipping is the best approach.
4. Extensive evaluation across different settings (continual and episodic), 5 types of adaptations, and types of corruptions.

## Weaknesses
1. Limited number of datasets and models. The paper only considers a single dataset (Imagenet-C) and a single model (ViT-Base/16). As done by [1], it would be interesting to see if the results transfer also to other datasets such as ImageNet-R or Imagenet-S, at least on a subset of the methods. Additionally, the evaluation is done only on ViT-Base/16.
2. While the empirical new insights on the use of non-private per-sample gradient clipping are interesting, I think the reason why it is the case is not fully explained. Is there any theoretical motivation for why per-sample gradient clipping is the best choice?
3. Overall, the originality of the TTA-DP methods seems quite incremental. The core idea follows DP-SGD. While Appendix A gives a detailed explanation of DP changes, most of the changes are due to DP-SGD or DP post-processing. Additionally, many of the commonly used techniques to further improve the performance of DP-SGD training recipe are not discussed (e.g., augmentation multiplicity, parameter averaging, or LoRA adaptations) [2]
4. The current evaluation focuses on standard clipping. It would be interesting to see whether other adaptive clipping methods would provide better trade-offs (e.g. [3]).

[1] “COME: Test-time adaption by conservatively minimizing entropy”. Zhang, Qingyang and Bian, Yatao and Kong, Xinke and Zhao, Peilin and Zhang, Changqing. ICLR 2025

[2] “Unlocking High-Accuracy Differentially Private Image Classification through Scale” Soham De, Leonard Berrada, Jamie Hayes, Samuel L. Smith, Borja Balle

[3] "Differentially private learning with adaptive clipping." Andrew, Galen, et al. NIPS 2021

---

> ### Author Rebuttal · Authors · 2026-03-31
>
> We thank the reviewer for their specific suggestions on DP and TTA in the constructive comments. We address the weaknesses and questions with new results and discussion for each point:
>
> > [more datasets and models] Limited number of datasets and models
>
> Thank you for the suggestion. We report rebuttal experiments on a new dataset, IN-R, and a new model from a different family, ConvNeXt. We report the accuracy of the baseline adaptation method and its clipping and DP variants at different privacy levels.
>
> ImageNet-R
>
> | method    | baseline | clip | $\epsilon=20$| $\epsilon=15$| $\epsilon=10$| $\epsilon=5$| $\epsilon=1$|
> |-----------|----------|------|-----:|-----:|-----:|-----:|-----:|
> | Tent      |     55.5 | 62.9 | 61.9 | 61.4 | 58.7 | 56.8 | 53.4 |
> | EATA      |     64.2 | 65.0 | 62.5 | 62.1 | 59.4 | 56.9 | 53.3 |
> | SAR       |     60.7 | 60.8 | 59.1 | 58.7 | 56.6 | 55.3 | 52.3 |
> | DeYO      |     62.9 | 63.7 | 59.2 | 58.7 | 57.0 | 55.1 | 51.9 |
> | DeYO_COME |     64.9 | 65.1 | 56.3 | 55.1 | 54.8 | 54.5 | 51.8 |
>
> ConvNeXT (continual, same setting as Tab. 1 and 8)
>
> | method | baseline | clip  | $\epsilon=20$| $\epsilon=15$| $\epsilon=10$| $\epsilon=5$| $\epsilon=1$|
> |-----------------|----------|------|-----:|-----:|-----:|-----:|-----:|
> | Tent            |     40.5 | 52.4 | 51.9 | 51.6 | 51.2 | 43.9 | 42.1 |
> | EATA            |     52.4 | 53.2 | 51.7 | 51.4 | 51.0 | 43.8 | 41.0 |
>
> ConvNeXT (episodic, 4 corruptions)
>
> | method | baseline | clip  | $\epsilon=20$| $\epsilon=15$| $\epsilon=10$| $\epsilon=5$| $\epsilon=1$|
> |-----------------|----------|------|-----:|-----:|-----:|-----:|-----:|
> | Tent            |     55.4 | 58.1 | 57.7 | 57.6 | 57.6 | 57.1 | 52.4 |
> | EATA            |     60.8 | 60.9 | 57.7 | 57.6 | 57.5 | 57.1 | 52.4 |
>
> Note that we evaluate methods in the continual adaptation setting (Tab. 1), and in an "express" version of the episodic setting (Tab. 9) on only a subset of the shifts (the last shift of each family) to finish during the rebuttal phase.
>
> While we focus on IN-C and ViT-B/16 for their popularity, we agree that more datasets and models provide breadth. We will incorporate these results into the appendix and point to them from Sec. 4.1 and 4.2.
>
> > [theory for per-sample clipping] any theoretical motivation for why per-sample gradient clipping is the best choice?
>
> The optimization for test-time adaptation is online, unsupervised, and over potentially shifted data. The inputs and losses may therefore vary greatly, resulting in gradients with highly different norms and directions. Per-sample clipping limits the influence of each input so none can dominate the aggregated update.
>
> Empirically, our ablations show that per-sample clipping performs better than both batch-level clipping and gradient-norm-based filtering. This suggests that controlling each sample’s contribution before aggregation is more effective than only clipping after aggregation or filtering samples using a hard threshold. However, theoretical justification of sample-level clipping for adaptation remains future work.
>
> > [more DP-SGD techniques] commonly used techniques to further improve the performance of DP-SGD training recipe are not discussed (e.g., augmentation multiplicity, parameter averaging, or LoRA adaptations) [2]
>
> We thank the reviewer for these helpful suggestions. With limited rebuttal time, we evaluate parameter averaging / EMA and augmentation multiplicity in a reduced setting: **DP-Tent**, under the **episodic setting**, on **four corruptions** only,.
>
> | $\epsilon$ | original | ema     | augmentation |
> |---------|----------|---------|--------------|
> |      20 |    65.9  | 66.3    | 66.2 |
> |      15 |    64.8  | 65.4    | 65.3 |
> |      10 |    62.0  | 63.1    | 63.1 |
> |       5 |    61.4  | 61.5    | 61.9 |
> |       1 |    54.9  | 54.3    | 54.8 |
>
> - For parameter averaging, we used EMA with the decay suggested in the paper (`ema_decay = 0.9999`) and several nearby values, while keeping the same hyperparameter tuning range as in our original experiments. As shown in the table, EMA gives modest improvements at larger privacy budgets, but the gain becomes marginal or disappears at smaller privacy budgets.
> - For augmentation multiplicity, we used horizontal flips and random crops as augmentations and computed the private gradient with 3 augmentations. As shown in the table, the performances slightly improves with an increased accuracy of $<1%$.
>
> > adaptive clipping [3]
>
> We thank the reviewer for this more sophisticated suggestion. Adaptive clipping is an interesting and relevant direction, and we will cite and discuss it in Section 2.3.
> We choose standard per-sample clipping because it is simple, widely used for DP, and provides a clean starting point for controlled comparisons across different TTA methods. We agree that adaptive clipping is a promising extension, and we view it as an important direction for future work.

---

> > ### Author Rebuttal · Reviewer_pR5g · 2026-04-02
> >
> > Thank you. My concerns have been adequately addressed.

---

> > > ### Author Response · Authors · 2026-04-07
> > >
> > > We thank the reviewer for their time and attention in reviewing this work and considering the rebuttal.

---

### Decision · Program_Chairs · 2026-04-30

**Decision:**

Accept (regular)

**Comment:**

This paper introduces a framework for Differentially Private Test-Time Adaptation by applying per-sample gradient clipping and Gaussian noise to model updates during inference to protect user data. Furthermore, the authors demonstrate that the per-sample clipping mechanism itself acts as a powerful regularizer that significantly improves the stability and accuracy of even standard, non-private adaptation methods.
While the proposed algorithm is very simple, it works very well and is clean. Reviewers also appreciated the original intersection of test-time adaptation with differential privacy, addressing a data-leakage vulnerability for machine learning models deployed in sensitive, real-world environments.  Overall, I recommend the paper for acceptance and believe the paper will make a strong contribution to ICML.